# *DICER1* hotspot mutation induces 3p microRNA gain of function via Argonaute strand switch

Sharan Malagobadan [1], Chunmei Shi[1], Acong Yang [1], Indranil Mondal[1], Habikah Baldeh[1], Karrie Spain[1], Wilfried Guiblet[2] & Shuo Gu [1] ✉

Dicer has RNase activity and is an essential enzyme in microRNA (miRNA) biogenesis. Mutations in the *DICER1* gene have been linked to various cancers, notably *DICER1* syndrome. To investigate the impact of pathogenic hotspot mutations in *DICER1*-associated tumors, we introduced a hotspot mutation into the endogenous *Dicer1* locus of a mouse embryonic carcinoma cell line using CRISPR. Our findings not only confirm the loss of 5p-miRNAs, as previously reported, but also demonstrate unexpected upregulation of specific 3p-miRNAs. These upregulated 3p-miRNAs, which are usually considered to be passenger strands in wild-type cells, are selectively loaded into the Argonaute protein in mutant cells based on their 5′-end characteristics, resulting in a 'strand-switch' phenomenon. Functional assays and transcriptome analyses demonstrate the activity of the passenger 3p-miRNAs. These results suggest that the Dicer hotspot mutation is not merely a loss-of-function mutation for 5p-miRNAs but also a gain-of-function mutation for passenger 3p-miRNAs, potentially contributing to *DICER1*-associated tumorigenesis.

MicroRNAs (miRNAs) are a class of short noncoding RNAs (~21–23 nt) that posttranscriptionally regulate most genes[1,2]. They join Argonaute (AGO) protein to form the RNA-induced silencing complex (RISC), which downregulates gene expression via mRNA degradation or translational inhibition. In animals, miRNAs recognize their targets through partial complementarity, allowing a single miRNA to regulate multiple genes[3]. Conversely, a single gene can be targeted by multiple miRNAs. These interactions mean that miRNAs function as master regulators of gene expression.

The biogenesis of a miRNA begins in the nucleus, where primary miRNA transcripts containing hairpin structures are recognized and processed by nuclear RNase III enzyme Drosha[4–6], generating hairpin precursor miRNAs. These precursors are exported into the cytoplasm, where Dicer cleaves the terminal loop to produce duplexes with 2-nt 3′ overhangs[7]. The miRNA duplex subsequently associates with the AGO protein to form an intermediate complex called pre-RISC[8]. During RISC maturation, one strand of the duplex is selected as the guide strand, while the other, the passenger strand, is ejected and degraded. The guide miRNA strand, embedded within AGO[9,10], directs the mature RISC to its target mRNA, primarily through seed sequence complementarity to the 3′ untranslated region (UTR)[11,12].

In the biogenesis pathway, the conversion of a precursor miRNA into a duplex is facilitated by Dicer's two RNase III domains, RNase IIIa and IIIb. These catalytic domains work in tandem, with RNase IIIa cleaving the 3p-arm and RNase IIIb cleaving the 5p-arm of the miRNA precursor, thereby liberating the miRNA duplex from the terminal loop[13]. The strand derived from the 5p-arm is known as the 5p-miRNA, whereas the strand from the 3p-arm is known as the 3p-miRNA (Fig. 1a). Based on 5′-end features (that is, thermodynamic stability and nucleotide identity)[14–16], either the 5p-miRNA or 3p-miRNA of each duplex will be preferentially loaded into AGO as the functional guide strand, while the other strand is discarded. This is known as AGO strand selection.

[1]RNA Mediated Gene Regulation Section, RNA Biology Laboratory, Center for Cancer Research, National Cancer Institute, Frederick, MD, USA. [2]Advanced Biomedical Computational Science, Frederick National Laboratory for Cancer Research, Frederick, MD, USA. ✉e-mail: shuo.gu@nih.gov

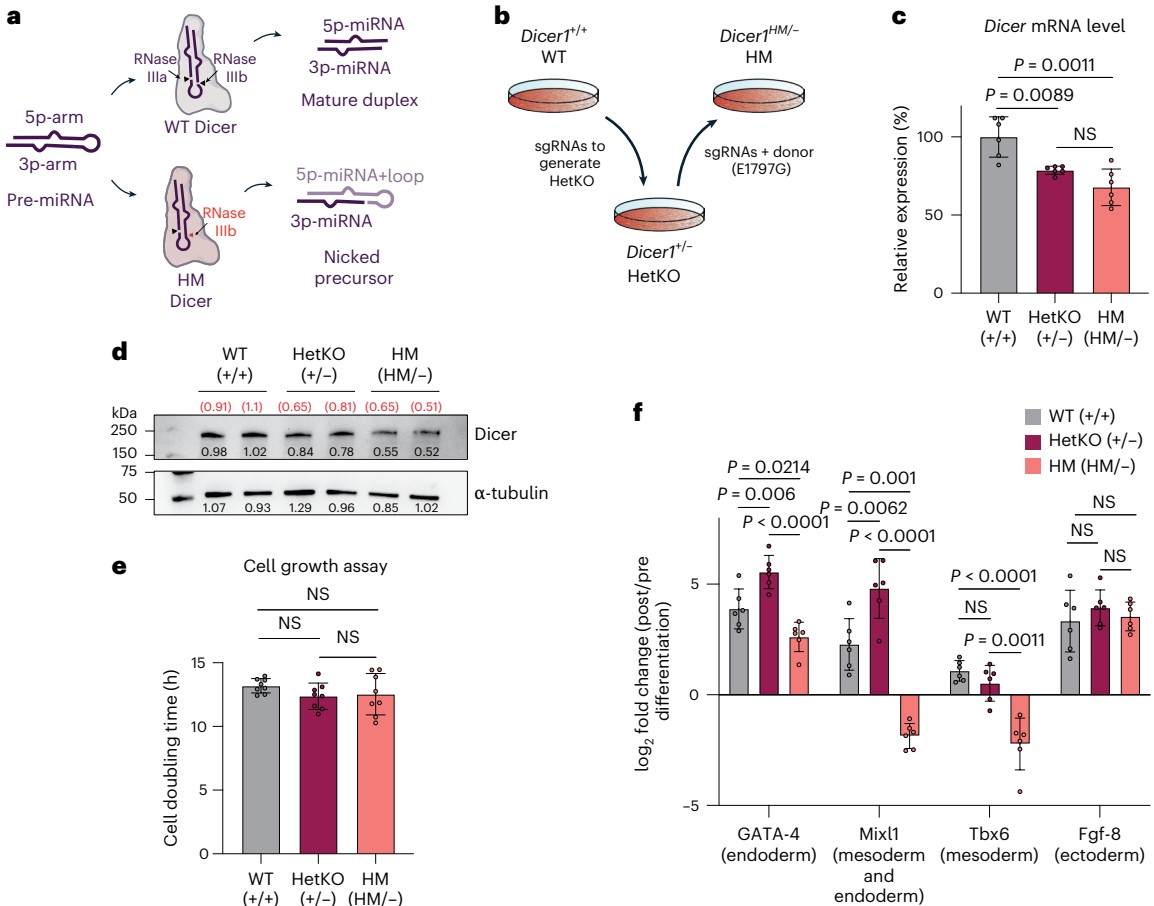

**Fig. 1 | *DICER1* tumor isogenic cell line was established with the P19 embryonic carcinoma cell line. a**, WT Dicer processes the hairpin miRNA precursor to generate a miRNA duplex consisting of 5p-miRNAs and 3p-miRNAs. By contrast, HM Dicer generates a nicked precursor with the 5p-miRNA still attached to the terminal loop. **b**, Generation of HetKO and HM cells from P19 embryonic carcinoma cells. P19 cells with WT Dicer were genetically modified with CRISPR to knock out one allele (HetKO). After a clonal cell line had been established, the remaining allele was targeted with allele-specific sgRNA to knock in the hotspot mutation (HM). **c**, Evaluation of Dicer mRNA levels by RT–qPCR (*n* = 6 replicates per cell line). **d**, Evaluation of Dicer protein expression by western blot, with each cell line run as duplicates across two independent experiments. Band intensity is denoted below each band (black font), and the normalized intensity for Dicer is denoted in parentheses (red font). **e**, Comparison of cell growth of WT, HetKO and HM cells (*n* = 8 replicates per cell line). **f**, RT–qPCR of markers of the three germ layers (shown in parentheses) after induction of differentiation using the hanging drop method. Fold change was calculated relative to the predifferentiation level for each cell line (*n* = 6 replicates per cell line). In **c**, **e** and **f**, two-sided Welch's *t*-test was used to calculate the *P* values, and data are shown as the mean ± standard deviation. NS, not significant.

Unlike *Drosophila* AGO paralogues that require additional cofactors, mammalian AGO has been shown to perform miRNA strand selection directly by sensing the 5' ends of miRNA duplexes using its MID domain[8,16].

Dicer mutations contribute to cancer progression, as in *DICER1* syndrome, a tumor predisposition disorder caused by impaired miRNA biogenesis[17,18]. *DICER1*-associated tumors encompass primarily pediatric neoplasms of the lung, kidney and thyroid, although other manifestations have also been characterized, including those affecting the female reproductive system[18]. Unique biallelic mutations of the *DICER1* gene underlie these tumors[19]. Specifically, a germline loss-of-function (LOF) mutation (first hit) with a somatic hotspot mutation (second hit) characterizes *DICER1*-associated tumors. In a study of pleuropulmonary blastoma, a pediatric lung cancer, deleterious Dicer germline mutations were observed in nearly 70% of patients[20]. Most hotspot mutations were pinned to the RNase IIIb domain at specific residues essential for RNase IIIb catalytic activity[21]. As a result, the mutant Dicer cleaves the 3p-arm (via functional RNase IIIa) but not the 5p-arm, leaving the 5p-miRNA attached to the terminal loop to be eventually degraded. Consequently, the hotspot mutations lead to loss of 5p-miRNAs and retention of 3p-miRNAs[21–27] (Fig. 1a). Notably,

a recently identified hotspot mutation in the RNase IIIa domain was found to still inhibit the function of RNase IIIb and impair production of 5p-miRNAs but not 3p-miRNAs[28]. The functional convergence of these hotspot mutations on the *DICER1* gene raises intriguing questions regarding the need for simultaneous retention of 3p-miRNAs and loss of 5p-miRNAs for tumorigenesis to occur.

Understanding the pathogenesis of hotspot mutations requires thorough investigation of their effects on miRNA biogenesis. Although the loss of 5p-miRNAs has been well documented, recent studies have begun to reveal dysregulation of 3p-miRNAs in *DICER1*-associated tumors[28–32]. However, whether this dysregulation is a direct consequence of the hotspot mutation and whether it has functional implications for tumorigenesis remain unclear. To explore the functional impact of the *DICER1* RNase IIIb mutation, we generated a set of cells mimicking *DICER1*-associated pleuropulmonary blastoma tumors using CRISPR. Our findings indicated that the *DICER1* RNase IIIb mutation leads to dysregulation in AGO strand selection, resulting in upregulation of a subset of 3p-miRNAs. Thus, the *DICER1* RNase IIIb hotspot mutation not only causes LOF of 5p-miRNAs, as previously reported, but also leads to gain of function (GOF) for certain 3p-miRNAs, potentially contributing to the pathogenicity of *DICER1*-associated tumors.

## Results

### *DICER1* tumor isogenic cell line was established with the P19 embryonic carcinoma cell line

Previous studies characterizing RNase IIIb mutation have relied on overexpression of mutant Dicer constructs in a Dicer-null background[23–26]. To model the *DICER1*-associated tumor genotype in a more physiological setting, we introduced a pathogenic RNase IIIb mutation into the endogenous *Dicer1* gene using the P19 cell line, a diploid mouse embryonic carcinoma cell line. This approach preserves Dicer levels similar to those in *DICER1* tumors. We first generated a *Dicer1* heterozygous knockout (KO; HetKO) using single guide RNAs (sgRNAs) to produce a frameshift truncation on one allele (Fig. 1b and Extended Data Fig. 1a). To confirm LOF, we cloned the truncated complementary DNA into an expression vector and overexpressed it in a HEK293T *DICER* KO cell line (Extended Data Fig. 1b). Northern blot analysis showed that unlike the wild-type (WT) Dicer, the truncated form failed to rescue either 5p-miRNA or 3p-miRNA production, confirming its LOF (Extended Data Fig. 1c). The remaining allele was then edited using CRISPR to introduce a hotspot mutation (E1797G) corresponding to human DICER1 E1813 (Extended Data Fig. 1d), the most frequently mutated RNase IIIb residue in *DICER1*-associated tumors[18,33]. The resulting hotspot mutant (HM) line harbored both a LOF and a hotspot allele, mimicking the genotype of *DICER1*-associated tumors (Fig. 1b).

Following confirmation of the mutation by Sanger sequencing (Extended Data Fig. 1e), we assessed Dicer expression by quantitative PCR with reverse transcription (RT–qPCR) (Fig. 1c) and western blotting (Fig. 1d). Both mRNA and protein levels of Dicer in HetKO and HM cells were reduced but remained higher than half of the levels in WT cells, as expected from monoallelic expression, suggesting a compensation mechanism to maintain Dicer expression. The mRNA and protein levels of HetKO and HM cells were comparable, indicating that the hotspot mutation itself does not substantially affect the stability of Dicer mRNA or protein.

Although previous studies have reported increased proliferation in HM cells[25], our growth assay showed that HM cells proliferated similarly to WT and HetKO cells (Fig. 1e and Extended Data Fig. 1f). This discrepancy was likely due to the different cell type used in this study, as P19 is an embryonic carcinoma cell line with a high proliferative rate and self-renewal ability. Similar to WT P19 cells, HM cells formed spheroids upon dimethyl sulfoxide (DMSO) treatment, indicating that the hotspot mutation alone does not eliminate the ability of cells to differentiate (Extended Data Fig. 1g). Nonetheless, HM but not HetKO cells manifested minor differential defects, as HM cells were able to express some germline markers, such as GATA-4 (endoderm) and Fgf-8 (ectoderm), but not others, such as Mixl1 (endoderm and mesoderm) and Tbx6 (mesoderm), upon differentiation (Fig. 1f). Although these markers alone do not provide a complete picture of how differentiation is impaired in HM cells, the inability to express certain differentiation markers may explain the pathogenic phenotype of the hotspot mutation; this warrants separate investigation. We concluded that the P19-based isogenic cell line closely mimicked cells with the *DICER1* pathogenic mutation and therefore used it for further mechanistic studies to examine the impact of the mutation on miRNA biogenesis.

### Expression of 5p-miRNAs is abolished in HM cells

After establishing and characterizing the cell lines, we performed miRNA sequencing (miR-seq) to examine the global impact of the RNase IIIb mutation on miRNA biogenesis. To enable direct comparison of samples with reduced Dicer dosage (HetKO and HM) to WT samples, spike-in was added during library preparation for normalization. miRNA levels were largely comparable between WT and HetKO (Fig. 2a and Extended Data Fig. 2a), suggesting that loss of one *Dicer1* allele did not affect overall miRNA expression. This was presumably owing to compensatory mechanisms when Dicer levels are reduced or the lack

of a bottleneck effect of Dicer on miRNA production. Either way, this result suggests that the established role of Dicer as a haploinsufficient tumor suppressor[18,34–36] is likely to be independent of its function in miRNA biogenesis.

Consistent with previous studies[23–27], HM cells showed a dramatic loss of 5p-miRNAs (Fig. 2a,b). Notably, small numbers of 5p-miRNAs were still detected in HM cells (Fig. 2b and Extended Data Fig. 2b), suggesting residual cleavage at the 5p-arm despite the RNase IIIb mutation. Nonetheless, sequencing and northern blot results showed that 5p-miRNAs, whether guide or passenger strands, were significantly downregulated in HM cells (Fig. 2c,d and Extended Data Fig. 2c).

We also observed an increase in long reads representing 5p+loop sequences for certain miRNAs in HM cells (Fig. 2b), supporting the notion that the RNase IIIb mutant generates nicked precursors. Northern blot results (Fig. 2d) confirmed loss of 5p-miRNAs but showed that not all 5p+loop intermediates were captured by next-generation sequencing. For instance, whereas miR-15b-5p and miR-20b-5p showed detectable intermediates in the northern blot, they were not evident in the miR-seq results (Fig. 2d and Extended Data Fig. 2d). This discrepancy was likely because the secondary structure of the 5p+loop intermediates impeded efficient library construction for some miRNAs. Although the selective detection of 5p+loop intermediates was intriguing, we did not pursue this further owing to sequencing-related technical limitations.

### A subset of passenger 3p-miRNAs is upregulated in HM cells

Whereas 5p-miRNAs were markedly downregulated, 3p-miRNAs as a whole remained at the same level in HM cells (Figs. 2a and 3a), consistent with previous characterizations of the impact of Dicer RNase IIIb mutations. However, closer examination revealed that a subset of 3p-miRNAs was upregulated (Fig. 3b and Extended Data Fig. 3a). miR-seq analyses indicated 4–10-fold upregulation of miR-7a-1-3p, miR-15b-3p, miR-17-3p and miR-20b-3p, among others. Notably, nearly all upregulated 3p-miRNAs were passenger strands, suggesting that the upregulation was unlikely to be random. In fact, the average level of passenger but not guide 3p-miRNAs increased specifically in HM cells (Fig. 3c), demonstrating that the upregulation induced by Dicer RNase IIIb mutation was specific to passenger strands. Northern blotting results confirmed this upregulation and the striking difference between the passenger and guide 3p-miRNAs (Fig. 3b,d), indicating that these findings were not artifacts of sequencing library construction.

Although all upregulated 3p-miRNAs were passenger strands, not all passenger 3p-miRNAs showed increased levels in HM cells. Among those that were upregulated, the extent of upregulation varied. The fold change of passenger 3p-miRNAs was independent of their abundance in WT cells (Extended Data Fig. 3b), suggesting that the observed upregulation in HM cells was not merely a consequence of low initial expression levels in WT cells. In addition, guide 3p-miRNAs expressed at levels comparable to those of passenger 3p-miRNAs in WT cells were not upregulated in HM cells (Extended Data Fig. 3a). As the absolute levels of all 3p-miRNAs did not differ significantly between WT and HM cells (Figs. 2a and 3a), it is also unlikely that the observed upregulation was a compensatory response to the loss of 5p-miRNAs.

Comparison with datasets from patients with *DICER1* tumors[30] and genetically engineered mouse models (GEMM) bearing the RNase IIIb hotspot mutations[31,32] revealed similar upregulation of passenger strands (Fig. 3e). Notably, 5p-miRNAs were still detectable in high abundance in these datasets (Extended Data Fig. 3c). Despite the inherent noise from sampling heterogeneous tumor cell populations, passenger 3p-miRNA upregulation persisted in both the patient and GEMM datasets (Fig. 3e). These findings strongly suggest that this upregulation is specific and driven by a hitherto unknown mechanism linked to the Dicer hotspot mutation.

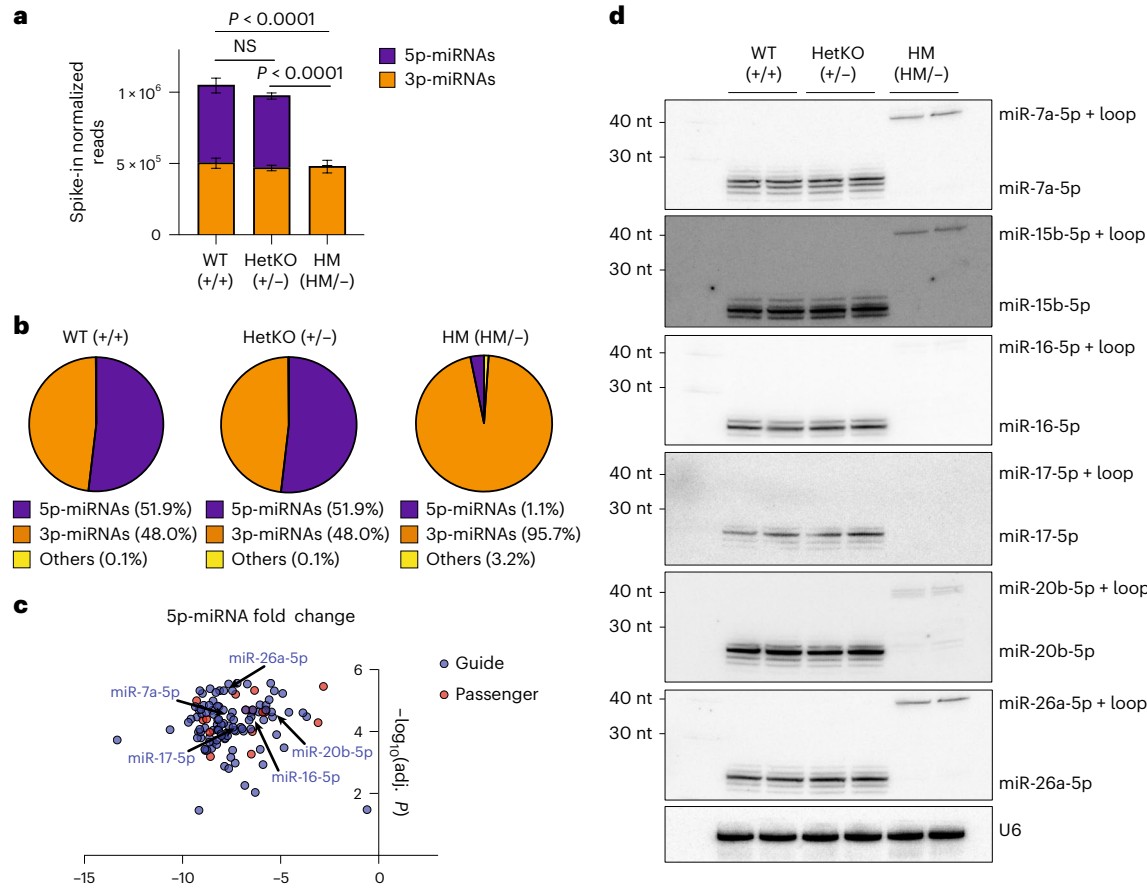

**Fig. 2 | Expression of 5p-miRNAs is abolished in HM cells. a**, Comparison of spike-in normalized reads of 5p-miRNAs and 3p-miRNAs in WT, HetKO and HM cells (*n* = 6 replicates per cell line). Two-sided Welch's *t*-test was used to calculate *P* values for total miRNA reads. Data are shown as the mean ± standard deviation. **b**, Breakdown of specific miRNA populations in each cell line. 'Others' denotes long reads for 5p-miRNAs (5p-miRNA+loop). **c**, Volcano plot for 5p-miRNA fold change against statistical significance. Guide (blue dots) and passenger (red dots) miRNAs were classified based on their expression relative to that of their complementary miRNA, where guide is the predominant strand of a miRNA duplex based on abundance (>70%) in WT cells. Two-sided Welch's *t*-test was used to calculate *P* values, followed by false discovery rate adjustment. **d**, Evaluation of miRNA expression by northern blotting using total RNA for each cell line. The blot was probed for 5p-miRNAs abundant in the WT cells. Adj., adjusted.

## Dysregulation of AGO loading results in passenger 3p-miRNA GOF

To understand the mechanism of passenger 3p-miRNA upregulation, we performed Dicer and AGO immunoprecipitation (IP), followed by northern blotting. In HM cells, miR-7a-1-3p, miR-17-3p and miR-20b-3p were associated with AGO but not Dicer (Fig. 4a). This suggests these upregulated passenger miRNAs are loaded into AGO rather than trapped in mutant Dicer. Notably, miR-26a-2-3p was an exception, being associated with Dicer but not AGO for an unknown reason. Nonetheless, these results indicates that the absence of miR-7a-1-3p, miR-17-3p and miR-20b-3p in Dicer IP was not due to inefficient IP. Next, we examined whether these upregulated 3p-miRNAs were functional. We generated a set of dual luciferase reporters bearing miRNA target sites in their 3' UTR as miRNA function sensors. The miRNA repression ability was measured by comparing the luciferase activity of the sensor to that of a seed mutant control. Whereas miR-7a-1-3p and miR-17-3p showed marginal repression ability in WT cells, their repression ability dramatically increased in the HM cells (Fig. 4b). As expected, miR-26a-2-3p, trapped in mutant Dicer, did not show repression in HM cells (Fig. 4b). These results suggest that a subset of passenger 3p-miRNAs are not only upregulated but gain functional activity in HM cells.

Our findings suggest that certain passenger 3p-miRNAs in WT cells were selected as the functional guide strand in HM cells.

Their stability increased owing to AGO protection, which at least in part accounted for their upregulation. This raised the question of how the Dicer RNase IIIb mutation led to dysregulation of strand selection, which is mediated by AGO proteins during RISC maturation, a step occurring downstream of Dicer processing. We hypothesized that AGO needs access to both the 5' and 3' ends of the guide strand for unwinding. In HM cells, AGO must load nicked hairpin precursors instead of miRNA duplexes. When the guide strand is in the 3p-arm, as with miR-92a, AGO can access both ends, loading the 3p-strand normally. However, if the guide strand is in the 5p-arm, as with miR-7a-1, AGO cannot access the 3' end owing to the attached loop, preventing unwinding. AGO would likely dissociate and rebind the nicked precursor in the opposite direction, allowing the passenger 3p-strand to become the guide strand (Extended Data Fig. 4a).

To test this model, we performed an in vitro RISC formation assay. To ensure the observed effect was specific to AGO, we incubated cell lysate from *DICER* KO cells with substrates mimicking WT or HM Dicer products. The functional mature RISC was trapped with the target oligo and visualized using a native gel system[37,38] (Extended Data Fig. 4b). We first performed the assay with the mir-92a duplex, radiolabeling either the 3p-strand or the 5p-strand. The mature RISC–target complex was detected only in the presence of the target oligo and exclusively with the guide 3p-strand, validating our assay (Fig. 4c and Extended

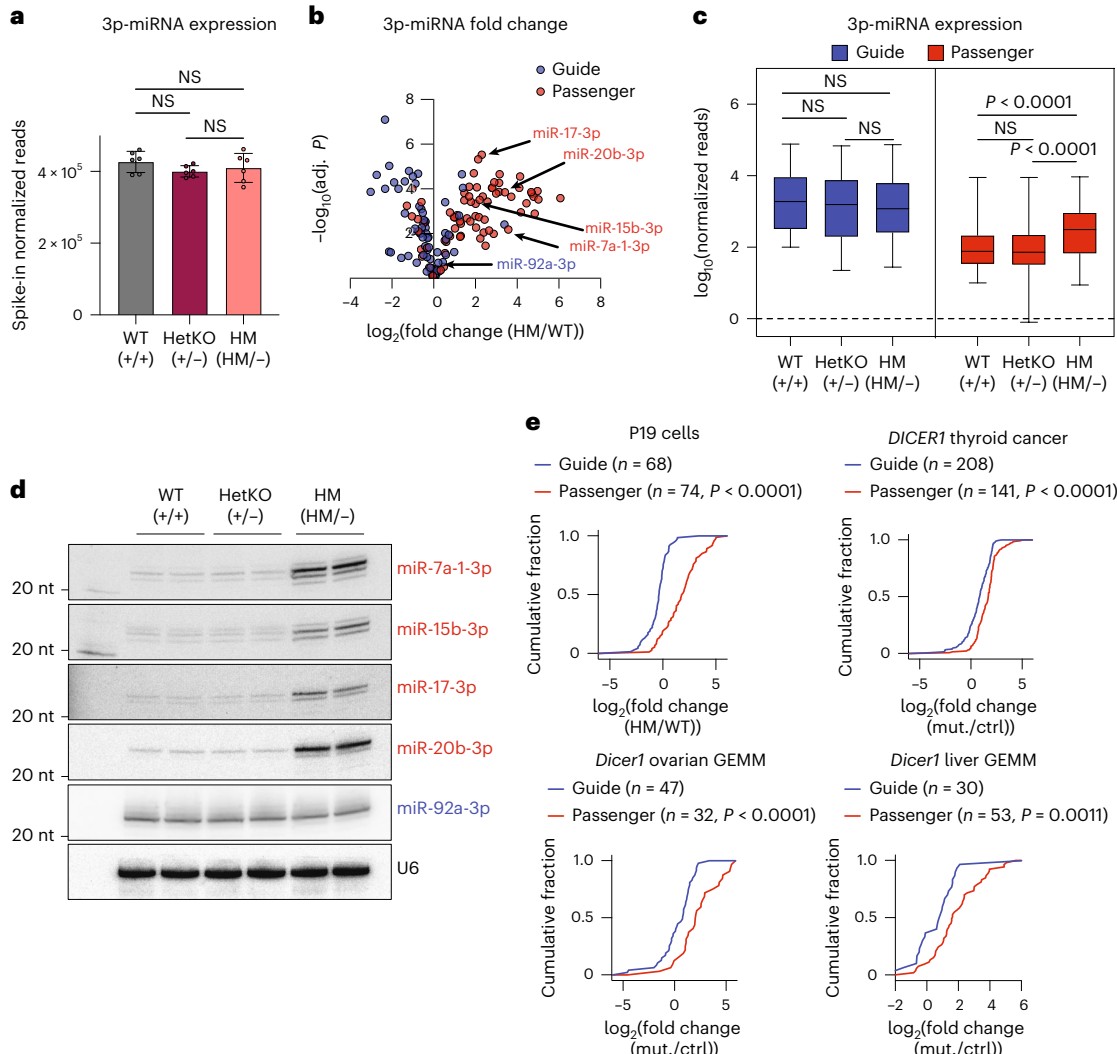

**Fig. 3 | A subset of passenger 3p-miRNAs is upregulated in HM cells.**
**a**, Comparison of spike-in normalized reads of 3p-miRNAs in WT, HetKO and HM cells ($n = 6$ replicates per cell line). Two-sided Welch's $t$-test was used to calculate $P$ values. Data are shown as mean ± standard deviation. **b**, Volcano plot for guide (blue dots) and passenger (red dots) 3p-miRNA fold change against statistical significance. Two-sided Welch's $t$-test was used to calculate the $P$ values, followed by false discovery rate adjustment. **c**, Box plot of miRNA expression shown as normalized reads for guide ($n = 69$) and passenger ($n = 94$) 3p-miRNAs in each cell line. Average reads from $n = 6$ replicates per cell line were used. Two-sided Welch's $t$-test was used to calculate the $P$ values. The box plot shows the median (center line), 25th–75th percentile (box) and minimum to maximum values (whiskers). **d**, Evaluation of miRNA expression by northern blot using total RNA for each cell line. The blot was probed for upregulated passenger 3p-miRNAs (miR-7a-1-3p, miR-15b-3p, miR-17-3p and miR-20b-3p). Guide 3p-miRNA (miR-92a-3p) is shown for comparison. U6 is shown as the loading control. **e**, Cumulative distribution of guide and passenger 3p-miRNA fold change in HM Dicer samples in P19 cells, *Dicer1* ovarian GEMM, *Dicer1* liver GEMM and the *DICER1* thyroid cancer dataset. One-sided Kolmogorov–Smirnov test was used to calculate the $P$ values. Mut., mutated; ctrl, control.

Data Fig. 4c). When using a nicked precursor mimicking mutant Dicer products, we observed a similar result, indicating that AGO loading of guide 3p-miRNAs was unaffected in HM cells (Fig. 4c). By contrast, results differed for the miR-7a-1 passenger 3p-strand when comparing duplex and nicked precursor substrates. With the duplex, AGO predominantly loaded the 5p-strand (Fig. 4d, comparing gel ii to gel i). However, when the nicked precursor was used, 5p+loop strand loading was completely abolished (Fig. 4d, comparing gel iv to gel ii), supporting our hypothesis that the loop structure at the 3′ end of the 5p-strand hinders its loading into AGO. Instead, the 3p-strand was preferentially selected as the guide (Fig. 4d, gel iii). These findings demonstrate that in Dicer mutant cells, 3p passenger strands can be loaded into AGO as functional guide strands. Notably, miR-7a-1-3p showed some ability to form mature RISC even from the duplex form (Fig. 4d, gel i), suggesting that miR-7a-1-3p was not exclusively selected as a passenger strand in the WT cells. Nonetheless, the 3p-strand formed mature RISC more

efficiently from a nicked precursor (Fig. 4d, comparing gel iii to gel i), phenocopying the increased AGO enrichment in HM cells.

To further test our model, we sought to determine whether precursors with a nick on the opposite strand would trigger a similar strand-switching phenomenon and lead to upregulation of the opposite strand. To this end, we generated a Dicer RNase IIIa mutant (D1320A) that produced precursors with a nick on the 5′ arm (Fig. 4e) and expressed it in a *Dicer*-null cell line. WT Dicer, the RNase IIIb mutant and a catalytically inactive Dicer served as controls (Extended Data Fig. 4d). The impact of this mutant on miRNA biogenesis was examined using northern blotting and small RNA sequencing. Consistent with previous findings[23], production of 3p-strands was completely abolished (Fig. 4f and Extended Data Fig. 4e), likely because the loop structure attached to the 5′ end of the 3p-strand interfered with recognition by the AGO MID domain. Notably, we did not observe consistent upregulation of the 5p passenger strands (Fig. 4f and Extended Data Fig. 4f). Instead, most

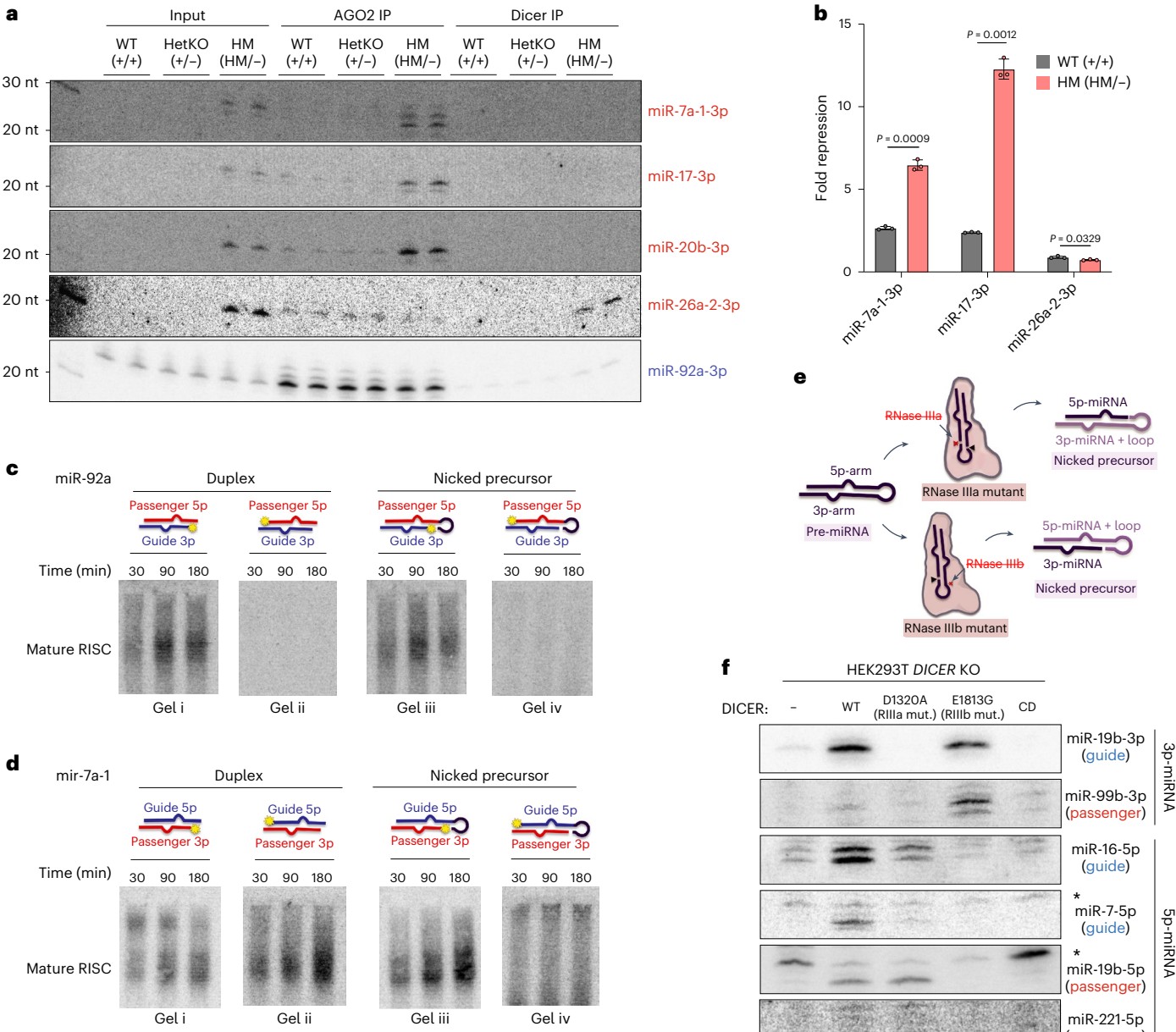

**Fig. 4 | Dysregulation of AGO loading results in passenger 3p-miRNA GOF.**
**a**, Evaluation of AGO2 and Dicer enrichment by northern blot using IP samples. The blot was probed for upregulated passenger 3p-miRNAs (miR-7a-1-3p, miR-17-3p, miR-20b-3p and miR-26a-2-3p). Guide 3p-miRNA (miR-92a-3p) is shown for comparison. **b**, Evaluation of miRNA function by reporter assay for upregulated passenger 3p-miRNAs (miR-7a-1-3p and miR-17-3p) in WT and HM cells ($n = 3$ replicates per cell line). AGO-depleted miR-26a-2-3p was included as a negative control. Fold repression was calculated by normalizing to the seed mutant reporter for each miRNA. Two-sided Welch's $t$-test was used to calculate the $P$ values. Data are shown as mean ± standard deviation. **c**,**d**, RISC formation

assay for miR-92a (guide 3p-miRNA; **c**) and miR-7a-1 (passenger 3p-miRNA; **d**) in duplex and nicked precursor forms to visualize mature RISC formation. The star on the duplex and nicked precursor illustration indicates the radiolabeled strand. **e**, Illustration of the different nicked precursors produced by RNase IIIa and IIIb. **f**, Northern blot of Dicer overexpression (WT, RNase IIIa mutant, RNase IIIb mutant and catalytically dead) in HEK293T *DICER* KO cells, probed for guide and passenger 5p-miRNAs and 3p-miRNAs (nonspecific bands are denoted by an asterisk). This blot is a qualitative validation of the miR-seq data presented in Extended Data Fig. 4e,f; no replicates were performed. RIIIamut., RNase IIIa mutant; RIIIbmut., RNase IIIb mutant; CD, catalytically dead.

5p-strands, regardless of whether they typically functioned as guide or passenger, were downregulated (Fig. 4f and Extended Data Fig. 4f). This suggests that although the loop portion was not covalently attached to the 5p-strand, it nonetheless obstructed access to its 3′ end. These results further support our hypothesis that AGO loading requires free access to both the 5′ and 3′ ends of the guide strand. Moreover, they may help explain why cancer-associated *DICER1* mutations predominantly affect RNase IIIb activity rather than RNase IIIa.

Overall, the in vitro and in vivo analyses implicate strand switching by AGO as the mechanism for the functional upregulation of passenger

3p-miRNAs. Thus, the Dicer hotspot mutation has downstream functional implications for AGO and serves as a GOF mutation for passenger 3p-miRNAs.

## The 5′ end features of upregulated passenger 3p-miRNAs determine their GOF

To systematically assess the GOF of passenger 3p-miRNAs, we compared their relative abundance in input samples versus AGO2 IP using miR-seq and calculated the enrichment of each miRNA in the AGO2 complex (Extended Data Fig. 5a). Passenger 3p-miRNAs in HM cells exhibited

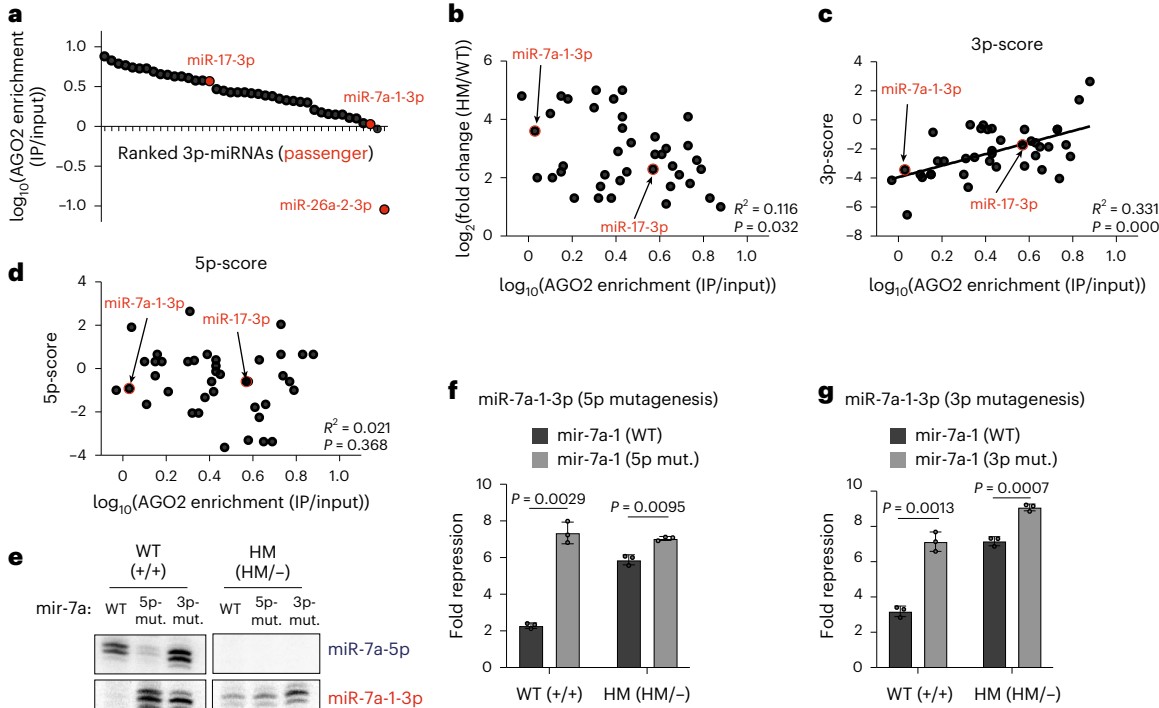

**Fig. 5 | The 5′ end features of upregulated passenger 3p-miRNAs determine their GOF. a**, Distribution of AGO2 enrichment ranked from most enriched to least. Enrichment was calculated as reads from AGO2 IP divided by reads from input (total RNA miR-seq). **b**, Plot of AGO2 enrichment versus fold change in total RNA miR-seq for upregulated passenger 3p-miRNAs. Pearson's correlation analysis was performed to determine $R^2$ and $P$ values. **c**,**d**, Plots of AGO2 enrichment versus 3p-score (**c**) and 5p-score (**d**) for upregulated passenger 3p-miRNAs. Pearson's correlation analysis was performed to determine $R^2$ and

$P$ values. **e**, Northern blot for mir-7a-1 mutagenesis. The 5p-strand was mutated to lower the 5p-score, whereas the 3p-strand was mutated to increase the 3p-score. **f**,**g**, Evaluation of miR-7a-1-3p function by reporter assay for 5p mutagenesis (**f**) and 3p mutagenesis (**g**) corresponding to the northern blot in **e**. Fold repression was calculated by normalizing to the seed mutant reporter for each miRNA in WT and HM cells ($n$ = 3 replicates per cell line). Two-sided Welch's $t$-test was used to calculate the $P$ values. Data are shown as mean ± standard deviation.

varying levels of enrichment (Fig. 5a), with higher enrichment indicating a greater likelihood of the miRNA being loaded into AGO as the functional strand. Consistently, the highly enriched miR-17-3p demonstrated more pronounced GOF in HM cells than the less enriched miR-7a-1-3p. By contrast, miR-26a-2-3p, which was trapped with Dicer, was depleted in AGO2 IP as expected (Figs. 4b and 5a). Thus, we used AGO enrichment as a proxy for functional activity in subsequent analyses. Notably, AGO enrichment of passenger 3p-miRNAs was not positively correlated with their upregulation levels in HM cells (Fig. 5b), suggesting that not all upregulated passenger strands were uniformly loaded as guide strands. Some of these upregulated 3p-miRNAs likely remained in the duplex form. Therefore, the extent of GOF for passenger 3p-miRNAs cannot be determined solely by their expression levels in HM cells.

To select the guide strand, AGO senses and compares the 5′ features of both strands, including thermostability and nucleotide composition. Suzuki et al. developed a mathematical model that combines these 5′ end features into a single score[16]. Guide strand selection typically involves competition between the two strands of a miRNA duplex, with the strand with the highest score being favored. Indeed, the difference between the scores of the 5p-miRNAs (5p-scores) and their 3p-miRNAs (3p-scores) (Extended Data Fig. 5b), rather than either score alone (Extended Data Fig. 5c,d), was most strongly correlated with strand selection, as reflected by the 5p/3p ratio in WT cells. Moreover, neither the 5p-score nor the 3p-score alone was correlated with AGO enrichment in WT cells (Extended Data Fig. 5e,f). By contrast, our model predicted that in HM cells, 3p-miRNA loading would depend solely on the features of the 3p-strand, as the loop attached to the 5p-miRNA would prevent its loading into AGO. Supporting this, AGO enrichment for passenger 3p-miRNAs was correlated with their 3p-scores (Fig. 5c) but not with the corresponding 5p-scores (Fig. 5d).

To test this prediction, we performed mutagenesis to modify the 5′ nucleotide of either 5p-miRNA or 3p-miRNA in miR-7a-1, thereby altering the 5p-score or 3p-score, respectively (Extended Data Fig. 5g). As expected, reducing the 5p-score improved the selection of 3p-miRNAs as the guide strand in WT cells, leading to 3p upregulation (Fig. 5e) and increased 3p-target repression (Fig. 5f). In HM cells, however, as the 5p-miRNA no longer competed for AGO loading, lowering the 5p-score had only a marginal impact on passenger 3p-miRNA expression and function (Fig. 5e,f). Similar results were obtained when altering the 5p-score of miR-17 (Extended Data Fig. 5h–j). By contrast, altering the 3p-score had clear effects on passenger 3p-miRNA expression and function in both WT and HM cells. Specifically, increasing the 3p-score of miR-7 led to upregulation (Fig. 5e) with increased target repression (Fig. 5g). Mutagenesis to lower the 3p-score of miR-17 (Extended Data Fig. 5h) had the opposite effect, resulting in downregulation of miR-17-3p (Extended Data Fig. 5i) and decreased 3p-target repression (Extended Data Fig. 5k). These results confirm that the 3p-score rather than the 5p-score influenced the functional upregulation of passenger 3p-miRNAs in HM cells. This also suggests that the most functional passenger 3p-miRNAs in HM cells are not simply those that are highly upregulated but those with the most favorable features for AGO strand selection.

## Upregulated passenger 3p-miRNAs have a functional impact on the transcriptome

Next, we assessed the effects of passenger 3p-miRNA GOF using our miR-seq and RNA-seq datasets. As miRNA expression (Extended Data Fig. 6a) and AGO2 IP (Extended Data Fig. 6b) data showed near-perfect concordance between WT and HetKO, we focused our functional analyses on comparisons between HM and WT. Unlike most studies,

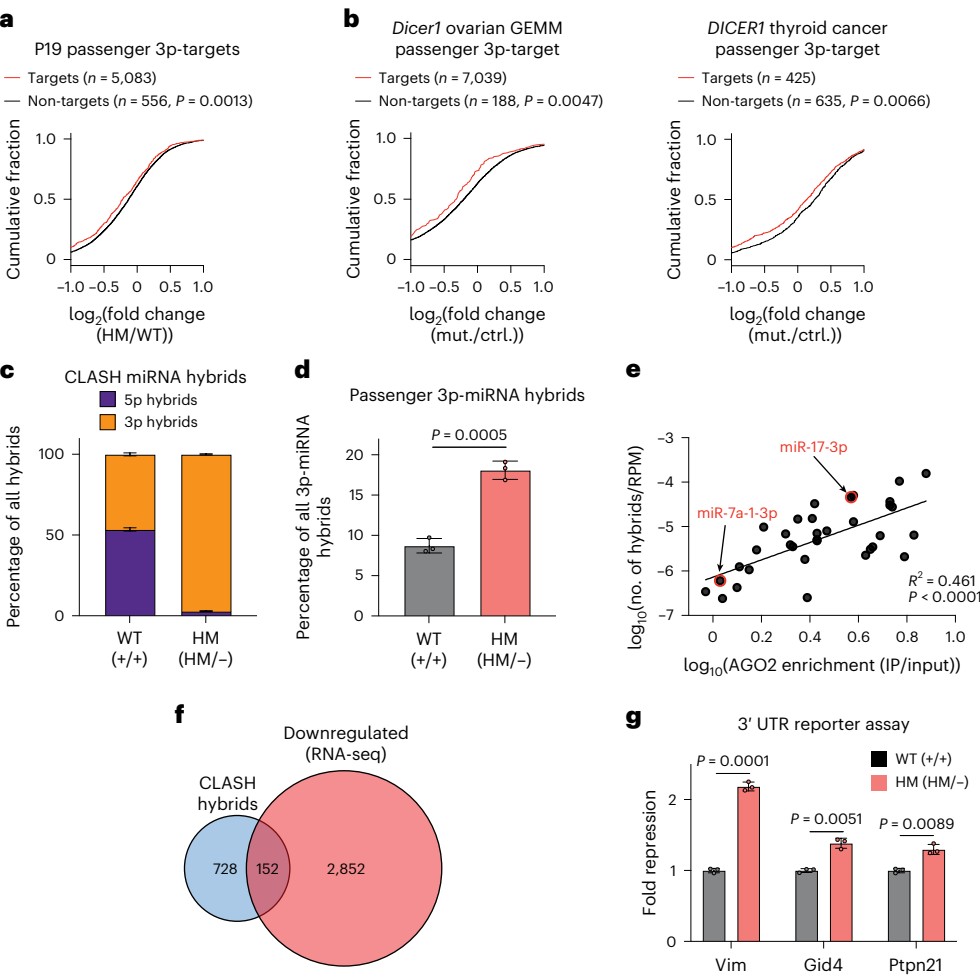

**Fig. 6 | Upregulated passenger 3p-miRNAs have functional impact on transcriptome. a,b**, Cumulative distribution of fold changes for predicted passenger 3p-targets in HM cells (**a**) and other matched miR-seq and RNA-seq datasets (*Dicer1* ovarian GEMM and patients with *DICER1* thyroid cancer) (**b**). One-sided Kolmogorov–Smirnov test was used to calculate the *P* values. **c**, Comparison of CLASH reads for 5p-miRNAs and 3p-miRNAs in WT and HM cells (*n* = 3 replicates per cell line). Data are shown as mean ± standard deviation. **d**, Comparison of CLASH reads for passenger 3p-miRNA hybrids in WT and HM cells shown as percentages of all 3p-miRNA hybrids (*n* = 3 replicates per cell line). Two-sided Welch's *t*-test was used to calculate the *P* values. Data are shown

as mean ± standard deviation. **e**, Plot of AGO2 enrichment versus number of passenger 3p-miRNA hybrids from CLASH (normalized to total RNA miR-seq expression). Pearson's correlation analysis was performed to determine the $R^2$ and *P* values. **f**, Venn diagram for targets from CLASH passenger 3p-miRNA hybrids and significantly downregulated genes from RNA-seq in HM cells. **g**, Reporter assay for passenger 3p-targets of miR-17-3p (Vim) and miR-30a-3p (Gid4 and Ptpn21) in WT and HM cells (*n* = 3 replicates per cell line). Fold repression was calculated by normalizing to the average fold change in WT cells. Two-sided Welch's *t*-test was used to calculate the *P* values. Data are shown as mean ± standard deviation.

in which only a small subset of miRNAs is dysregulated, our isogenic cell line presented a unique challenge, having not only gained passenger 3p-miRNAs but also lost approximately 50% of all miRNAs, leading to derepression of 5p-miRNA targets. To accurately interpret the true impact of passenger 3p-miRNAs amid these secondary perturbations, we grouped genes based on the predominant miRNA targeting them. Using TargetScan prediction[39,40], we categorized genes primarily targeted by endogenous 5p-miRNAs or passenger 3p-miRNAs as 5p-targets and passenger 3p-targets, respectively (Extended Data Fig. 6c). As expected, 5p-targets were specifically upregulated in HM cells (Extended Data Fig. 6d), validating our approach, whereas passenger 3p-targets were downregulated (Fig. 6a). We applied this analysis to other matched miR-seq and RNA-seq datasets (*Dicer1* ovarian GEMM[31] and patients with *DICER1* thyroid cancer[30]) and observed similar trends of upregulation of 5p-targets (Extended Data Fig. 6d) and downregulation of passenger 3p-targets (Fig. 6b), further supporting the GOF of passenger 3p-miRNAs.

Although the impact on 5p-targets and passenger 3p-targets in HM was statistically significant, the difference was marginal (Fig. 6a),

underscoring the limitations of a prediction-based approach. This is especially true for passenger 3p-miRNAs, as target predictions are generally optimized for the more broadly expressed guide strands. To provide more conclusive evidence for passenger 3p-miRNA GOF and simultaneously identify the relevant endogenous targets, we performed AGO2 cross-linking, ligation and sequencing of hybrids (CLASH) in both WT and HM cells. This biochemical assay captures actual miRNA–target interactions by ligating AGO-bound miRNAs to their target fragments, followed by high-throughput sequencing to generate hybrid reads. In WT cells, hybrid reads showed an equal distribution between 5p-miRNAs and 3p-miRNAs, whereas hybrids in HM cells were predominantly from 3p-miRNAs, as expected (Fig. 6c). Detailed analysis of the 3p-miRNA hybrids revealed an increased percentage of passenger 3p-miRNAs in HM cells (Fig. 6d), indicating GOF, as more passenger 3p-miRNAs bound to their targets in HM compared to WT cells. Notably, passenger 3p-miRNAs also showed a lower but appreciable number of hybrids in WT cells, indicating that they were not entirely nonfunctional in WT cells. Moreover, passenger 3p-miRNAs with greater AGO enrichment generated higher numbers of hybrid

reads, as shown by their positive correlation (Fig. 6e). This further demonstrated that the most functional passenger 3p-miRNAs were those with the greatest AGO enrichment. Gene ontology analysis of passenger 3p-targets revealed significant enrichment of transcription regulation (Extended Data Fig. 6e), indicating a potential role of passenger 3p-miRNAs in shaping the transcriptome. Future studies using isogenic human cell lines and patient-derived cells are needed to identify more disease-relevant targets.

Finally, we combined CLASH and RNA-seq data to identify and validate target genes regulated by the upregulated passenger 3p-miRNAs as a proof of concept. Consistent with the observation that passenger 3p-miRNA upregulation occurred in HM but not HetKO cells, downregulation of the corresponding target genes was specific to HM cells (Fig. 6f and Extended Data Fig. 6f). Among the 152 downregulated passenger 3p-targets, Vimentin (Vim), protein tyrosine phosphatase non-receptor type 21 (Ptpn21) and glucose-induced degradation protein 4 homolog (Gid4) were chosen for the whole 3′ UTR reporter assay. Vim is a predicted target of miR-17-3p, and Ptpn21 and Gid4 are targets of miR-30a-3p (Extended Data Fig. 6g). We evaluated target repression by cotransfecting the UTR reporters with the respective miRNA expression vectors and observed greater repression in HM cells compared to WT cells for all three targets (Fig. 6g). Hence, the target prediction, CLASH and reporter assays provided a consensus for passenger 3p-miRNA GOF.

## Discussion

Since the discovery of *DICER1* mutations in pleuropulmonary blastoma in 2009 (ref. 17), increasing efforts have been dedicated to understanding how these mutations affect miRNA biogenesis. Here, we found that the RNase IIIb hotspot mutation not only led to loss of 5p-miRNA expression but also induced gain of 3p-miRNA expression through a strand-switch mechanism. Notably, these 3p-miRNAs are considered to be passenger strands and are typically absent from WT cells. By employing target prediction, CLASH and reporter assays, we demonstrated that these upregulated passenger 3p-miRNAs were functional, as evidenced by their ability to load into AGO and repress their targets. This discovery provides mechanistic insight into the GOF effects associated with *DICER1* mutations and expands our understanding of miRNA biogenesis.

Our study further clarifies the functional consequences of the Dicer RNase IIIb hotspot mutation for 5p-strand production. Previous studies have shown that HM Dicer cannot generate 5p-miRNAs[23–26], leading to derepression of 5p-miRNA targets. For instance, upregulated let-7a-5p targets conferred oncogenic phenotypes on granulosa cells expressing mutant Dicer[26]. Although we detected small amounts of 5p-miRNAs in our isogenic line, the levels of these miRNAs were insufficient to suppress targets, as evidenced by the strong upregulation of the targets. Furthermore, while the mutant Dicer with a defective RNase IIIb domain is expected to produce nicked precursors (5p+loop and 3p), 5p+loop intermediates were detectable for only some miRNAs. Although the function of these intermediates remains unclear, their selective stability is intriguing. Understanding the basis of this stability may help us to evaluate 5p+loop intermediates as biomarkers for *DICER1*-associated tumors.

Our study demonstrates that the *DICER1* RNase IIIb hotspot mutation confers a GOF for passenger 3p-miRNAs, potentially contributing to tumorigenesis. These upregulated 3p-miRNAs may impair differentiation or inhibit tumor suppressor genes, promoting transformation. Notably, Vim, a key cytoskeletal component and a hallmark of mesenchymal cells, is a validated target of these 3p-miRNAs[41]. HM cells failed to express specific mesodermal markers upon DMSO-induced differentiation, indicating a potential defect in formation of mesenchymal lineage cells, which are thought to be the potential cell type of origin for *DICER1*-associated tumors[42]. These findings suggest that GOF in passenger 3p-miRNAs may critically drive tumorigenesis.

Moreover, our mechanistic insights support prediction and identification of functional passenger 3p-miRNAs, which will be valuable for future studies in clinically relevant contexts.

Notably, the upregulation of passenger 3p-miRNAs was also independently confirmed using human embryonic stem cells in a companion paper[43], demonstrating that the underlying mechanism is conserved between human and mouse. However, comprehensive identification of disease-relevant targets of passenger 3p-miRNAs requires deeper understanding of the specific contexts in which the hotspot mutation drives tumorigenesis, as miRNA and target gene expression vary across tissues and developmental stages. This represents a major limitation of our study, as the P19 cell line may not accurately reflect the tumor cell of origin. However, P19 cells are capable of differentiating into muscle lineage cells[44], the precursor of which (primitive mesenchymal cells) is considered to be a potential cell of origin of rhabdomyosarcoma[45], a known *DICER1*-associated tumor[18]. It would be intriguing to evaluate the growth and tumorigenic potential of this panel of isogenic P19 cells following DMSO-induced myogenic differentiation, both in vitro and in vivo. Such evaluation, if its results were validated by comparison of HetKO and HM cells, would enable rigorous investigation of the newly acquired targets of GOF 3p-miRNAs and help to identify the specific pathways affected. Furthermore, it will be important to examine how the RNase IIIb mutation affects the fidelity of RNase IIIa cleavage, potentially leading to the production of 5′ isomiRs from 3p-miRNAs and consequent changes to the transcriptome. Such future studies will be essential to fully elucidate the impact of RNase IIIb hotspot mutations and to understand why they are preferentially selected as the second somatic hit in tumors.

Our study offers insights into RISC maturation, which proceeds in two steps: loading of the miRNA duplex into AGO and subsequent strand unwinding. The first step is guided by the MID domain of AGO, which enforces strand selection based on its affinity for one strand of the miRNA duplex[16]. Unwinding removes the passenger strand to generate mature RISC and is initiated by N-domain wedging and facilitated by PAZ domain docking at the 3′ end[38,46]. Although strand selection has traditionally been attributed to 5′ end features, our findings reveal that 3′ end availability also plays a part, potentially overriding MID-mediated selection. In nicked precursors, the loop attached to the 5p-strand may obstruct N-domain wedging and/or PAZ docking, shifting strand selection toward the 3p-strand. Notably, the unwinding step can be bypassed when the passenger strand is cleaved by AGO slicing[8,46], eliminating the need for PAZ docking[46]. It would be intriguing to investigate how miRNAs with sliceable passenger strands are loaded into AGO in HM cells. Altogether, our study highlights *DICER1*-associated tumors as an example in which a pathogenic Dicer mutation alters AGO strand selection by modulating accessibility of the miRNA 3′ end.

Our findings have important implications for cancer therapeutics and diagnostics. Given the complex interactions between miRNAs and their targets, off-target effects pose a major challenge in miRNA-based therapies. The GOF by passenger 3p-miRNAs presents a unique opportunity: as these miRNAs are typically absent from normal cells, they represent ideal targets for antagomiRs with minimal off-target effects. In addition, passenger 3p-miRNAs show promise as biomarkers. Their presence, which is undetectable in normal cells, could serve as a diagnostic marker for somatic hotspot mutations before tumorigenesis. This approach may offer a faster alternative to sequencing for identification of biallelic *DICER1* mutations, enabling earlier intervention and improving patient outcomes.

## Online content

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

## Methods

### Cell culture

P19 and HEK293T cell lines were obtained from ATCC. The cells were cultured and maintained in DMEM (Gibco, 11965118) supplemented with 10% heat-inactivated fetal bovine serum (HyClone, SH30070.03HI) and 100 U ml$^{-1}$ penicillin–streptomycin (Gibco, 15140163). HetKO cells were generated using CRISPR with a pair of sgRNAs targeting exon 26 to generate LOF truncation in one allele of Dicer. HM cells were generated using another pair of sgRNAs specific to the full-length allele and donor DNA with the missense E1797G mutation. All the sgRNA and donor DNAs used were generated by the NCI LASP Genome Modification Core.

### RT–qPCR

Total RNA was extracted using TRIzol reagent (Invitrogen, 15596018). Reverse transcription was performed with 100 ng of total RNA using a SuperScript IV Reverse Transcriptase kit (Invitrogen, 18090010) and oligo (dT)12–18 primer to generate complementary DNA. qPCR was performed using iQ SYBR Green Supermix (Bio-Rad, 1708880). The expression of genes of interest was normalized to that of *GAPDH* using the ΔΔCq method. Fold changes for genes of interest were calculated by dividing postdifferentiation by predifferentiation expression for each sample.

### Western blot

Cells were lysed in modRIPA buffer (10 mM Tris pH 7.0, 150 mM NaCl, 1 mM EDTA, 1% Triton X-100 and 0.1% SDS) with cOmplete Mini Protease Inhibitor Cocktail (Roche, 11836153001). After quantification, 15 mg of each protein sample was run on a 4–20% Mini-PROTEAN TGX Stain-Free Gel (Bio-Rad, 4568093) and transferred using a Trans-Blot Turbo Transfer system onto a polyvinylidene fluoride membrane. Primary antibodies used in this study were rabbit anti-Dicer (1:1,000, Novus Biological, NBP1-06520), mouse anti-α-tubulin (1:5,000, Sigma, T9026) and rabbit anti-GAPDH (1:1,000, Cell Signaling Technology, 2118). Proteins were detected using Pierce ECL plus Western Blotting Substrate (Thermo Scientific, 32132) and imaged with a ChemiDoc imaging system.

### Cell growth assay

Cell growth was measured in real time using an xCELLigence Real-Time Cell Analysis instrument. For each replicate, a well of the xCELLigence E-plate (Agilent, 5469813001) was seeded with 5,000 cells, followed by incubation at 37 °C with 5% CO$_2$. The impedance was recorded at 15-min intervals for the duration of the assay.

### Northern blot

Total RNA was extracted using TRIzol reagent (Invitrogen, 15596018). After quantification, 20 mg of total RNA was run on a denaturing 20% (w/v) acrylamide gel with 8 M urea. A $^{32}$P-labeled Decade marker (Life Technologies, AM7778) was run alongside samples. The resolved RNA was transferred to a Hybond-N+ membrane (Cytiva, RPN303B) and ultraviolet (UV) cross-linked. $^{32}$P-labeled DNA probes against the miRNA of interest were hybridized overnight with the membrane in PerfectHyb Plus hybridization buffer (Sigma Aldrich, H7033) at 37 °C. The membranes were rinsed twice with 2× saline sodium citrate containing 0.1% SDS for 30 min at 37 °C. After the last rinse, the membrane was exposed to BAS Storage Phosphor Screen (Cytiva, 28956475) overnight and imaged using a Typhoon Trio imaging system. Image processing and band quantification were performed using ImageJ2 v. 2.14.0. Bands were cropped to include at least one marker band. The marker lane was omitted when the probed miRNA intensity obscured the marker signal during exposure (Fig. 4f) or when the gap between the band and its nearest marker was too big (Fig. 5e and Extended Data Fig. 5i). Source blot images provide relative references for marker positions.

### Differentiation assay

Cells were detached with trypsin and resuspended at a concentration of $1 \times 10^4$ cells per ml in a complete growth medium with 1% DMSO.

Droplets of 10 ml suspension were plated onto the inner surface of a 10-cm tissue culture plate lid. Upon plating, the plate was flipped to allow the cell droplets to hang off the lid and incubated at 37 °C. After 3 days, the spheroids from the hanging drops were transferred to a standard tissue culture plate to grow in adherent conditions for an additional 4 days before RNA extraction for RT–qPCR.

### miR-seq

Small RNA libraries for miR-seq were generated using a QIAseq miRNA Library Kit (Qiagen, 331502) according to the manufacturer's instructions. For normalization, 1 µl of QIAseq miRNA Library QC Spike-ins (Qiagen, 331535) was added to 500 ng of RNA before library preparation. The library was size-selected on a native 6% (w/v) acrylamide gel and purified using ethanol precipitation. The library was sequenced on an Illumina NextSeq 1000 platform.

### Immunoprecipitation

For endogenous AGO2 and Dicer IP, cells in a 10-cm dish were lysed in 1 ml modRIPA buffer (10 mM Tris-Cl pH 7.0, 150 mM NaCl, 1 mM EDTA, 1% Triton X-100 and 0.1% SDS) supplemented with cOmplete Mini Protease Inhibitor Cocktail (Roche, 11836153001). The cell lysate was split into two 500-µl portions. The two portions were incubated with 50 µl SureBeads Protein G Magnetic Beads (Bio-Rad, 161-4023) prepared with 5 µg mouse anti-AGO2 (WAKO, 014-22023) or anti-Dicer (Novus Biological, NBP1-06520) antibody at 4 °C overnight with rotation. After washing three times with Tris-buffered saline (50 mM Tris-HCl (pH 7.4), 150 mM NaCl) at room temperature, the beads were lysed in 1 ml TRIzol reagent (Invitrogen, 15596018) for RNA extraction. The RNA was used for northern blotting and miR-seq.

### RISC formation assay

The RISC formation assay was performed as previously described[37,38], with a modified lysate. The reactions were assembled in a 10 µl volume consisting of 4 µl lysate, 3 µl 40× reaction mix, 1 µl 15% (w/v) Ficoll 400 in lysis buffer, 1 µl 100 nM radiolabeled substrate, and 1 µl 100 nM 2′-O-methyl antisense oligonucleotide against the radiolabeled strand. The lysate used in the reactions was derived from HEK293T *DICER* KO cells transfected with FLAG-tagged AGO2 expression plasmid and collected after 24 h. The RNA substrates were synthesized with 5′ phosphate modification (Integrated DNA Technologies). After 5′ relabeling with $^{32}$P of either the 5p-strand or 3p-strand, the two strands were annealed by incubation at 95 °C for 2 min and 25 °C for 30 min to form substrates. The reactions were incubated at room temperature for the indicated amount of time and run on a native 1.5% agarose gel at 300 V for 90 min at 4 °C to resolve the mature RISC. The resolved RNA–protein complex was transferred to a Hybond-N+ membrane (Cytiva, RPN303B) and UV cross-linked. After overnight exposure to BAS Storage Phosphor Screen (Cytiva, 28956475), mature RISC with the radiolabeled strand was visualized using a Typhoon Trio imaging system. Images were processed using ImageJ2 v.2.14.0.

### Reporter assay

Dual luciferase reporter assays were performed using a psiCHECK-2 plasmid (Promega, C8021), in which the target sequence for miRNA of interest was cloned into the 3′ UTR of the Renilla luciferase. The plasmids were transfected with Lipofectamine 3000 (Invitrogen, L3000001) according to the manufacturer's instructions. For the endogenous miRNA reporter assay, a reporter plasmid with a fully complementary miRNA target sequence (2× perfect targets) was transfected (50 ng per well). For miRNA mutagenesis, miRNA expression vectors (100 ng per well) were cotransfected with the corresponding miRNA reporters (100 ng per well). For the whole 3′ UTR reporter assay, reporter plasmids with full-length 3′ UTRs of target genes (50 ng per well) were cotransfected with the corresponding miRNA expression vectors. For all reporters, luciferase activity was measured 24 h after

transfection. Renilla luciferase activity was internally normalized to that of firefly luciferase activity, and seed mutant reporters were used to calculate fold repression.

## RNA-seq

The library for mRNA-seq was generated using Illumina Stranded mRNA Prep (Illumina, 20040532) and sequenced on a NovaSeq 6000 SP using paired-end sequencing by the NCI CCR Sequencing Facility.

## CLASH

CLASH was used to identify miRNA–target RNA interaction sites using the iCLIP2 (ref. [47]) and CLASH[48] protocols as previously described. In brief, a 10-cm dish of confluent cells was irradiated with UV-C (300 mJ cm$^{-2}$) to covalently bond RNA-binding proteins to bound nucleic acids. During the subsequent cell lysis, the lysate was treated with Turbo DNase (Invitrogen, AM2238) to remove DNA and RNAse 1 (Thermo Scientific, EN0601) to partially digest the RNA and generate 25–200-nt fragments. IP of AGO2 proteins was performed using Dynabeads Protein G (Invitrogen, 10003D) conjugated with a mouse AGO2 antibody (WAKO, 014-22023) or IgG antibody for 4 h at 4 °C. After phosphorylation with T4 PNK (New England Biolabs, M0201), T4 RNA ligase 1 (New England Biolabs, M0204) was used for intermolecular ligation, followed by dephosphorylation and 3′ adapter ligation using T4 RNA ligase 2, truncated K227Q (New England Biolabs, M0351). T4 PNK labeling of RNA with $^{32}$P allowed visualization of the AGO2–RNA complexes after SDS polyacrylamide gel electrophoresis and transfer to a nitrocellulose membrane. The membrane sections containing protein–RNA complexes were cut and treated with proteinase K (Invitrogen, AM2542) to yield protein-free RNA. The 5′ ends were then phosphorylated to prepare the RNA for sequencing library preparation. The final AGO2 CLASH library was sequenced on the Illumina NextSeq 1000 platform using paired-end sequencing.

## Data analysis

RNA-seq data were processed using BBMerge[49] and RSEM[50] to merge the paired-end reads and quantify gene expression. miR-seq data were processed with QuagmiR[51] to quantify miRNA expression levels. For total RNA miR-seq data, reads were normalized to spike-in. For AGO2 IP miR-seq, reads were normalized to total miRNA. Datasets for *DICER1* thyroid cancer (BioProject ID: PRJNA934932), *Dicer1* ovarian GEMM (BioProject ID: PRJNA976189), and *Dicer1* liver GEMM (BioProject ID: PRJNA979254) were obtained from the Gene Expression Omnibus and analyzed similarly.

The cumulative distribution of miRNA fold changes was calculated using a custom R script. Guide and passenger miRNA classification was based on the relative percentages of the two miRNAs from the same duplex. The miRNA with >70% expression and >100 reads was designated as the guide strand, whereas the complementary strand was defined as the passenger strand. The thermodynamic energy for the first three nucleotides of each miRNA was calculated using RNAfold[52], RNAeval and a custom R script. The 5p-miRNA and 3p-miRNA scores were calculated using the asymmetry equation developed by Suzuki and colleagues[16]. Pearson's correlation analysis was performed using GraphPad Prism 10.

Target prediction was done using the predicted target list from TargetScan 8.0 (refs. [39,40]). A custom R script was used to filter for the top 10% of predicted targets based on context scores and to trim the top and bottom 25% of all expressed genes to exclude genes with excessively high or low expression. After filtering, the cumulative abundances of 5p-miRNAs, guide 3p-miRNAs and passenger 3p-miRNAs targeting each gene were calculated based on total RNA miR-seq reads. Genes predicted to be targeted by 5p-miRNAs more than 3p-miRNAs in WT cells were categorized as 5p-targets. Genes for which the gain in passenger 3p-miRNAs was greater than the loss of 5p-miRNAs in HM cells were categorized as passenger 3p-targets.

Gene ontology analysis was performed using the Enrichr web tool and GO Biological Process 2023 for targets from all passenger 3p-miRNA hybrids in HM. CLASH data were processed with QuagmiR to map the hybrid's miRNA fragment and Bowtie2 to map the target fragment to the 3′ UTR of mouse coding genes. To reduce false-positive hybrids, the resulting miRNA–target hybrid list was filtered to include only interactions predicted by TargetScan 8.0.

## Statistics and reproducibility

No statistical method was used to predetermine sample size. Sample sizes were selected based on the magnitude of effects observed and previous literature. Statistical analyses were performed using GraphPad Prism 10 for Pearson's correlation analysis and R software for two-sided Welch's *t*-test and one-sided Kolmogorov–Smirnov test. Data distribution was assumed to be normal, but this was not formally tested. Randomization was not relevant to this study. Samples were allocated based on genotype or treatment conditions. As data collection was instrument-based and quantitative, the risk of investigator bias was minimal. Each analysis was further supported by qualitative and orthogonal validation. miRNAs expressed from a chromosome 12 cluster were excluded from the results shown in Figs. 1–5. The chromosome 12 miRNA cluster was not expressed in the HM cell line owing to unexpected silencing. As the effect was transcriptional and clonal, these miRNAs were excluded to avoid confounding the posttranscriptional effect on Dicer processing and AGO enrichment. All attempts at replication were successful, and the numbers of replications are stated in the figure legends.

## Reporting summary

Further information on research design is available in the Nature Portfolio Reporting Summary linked to this article.

## Data availability

The sequencing datasets generated in this study (miR-seq, RNA-seq and CLASH) have been deposited in the Gene Expression Omnibus under accession codes GSE279556, GSE279645 and GSE279636. Oligonucleotide sequences are provided as Supplementary Information. Additional data that support the findings of this study are available via figshare at https://doi.org/10.6084/m9.figshare.29598575 (ref. [53]). All other data supporting the findings of this study are available from the corresponding author on reasonable request. Source data are provided with this paper.

## Code availability

Code used for data analysis can be found at the GitHub repository for the cumulative distribution of guide and passenger 3p-miRNA fold change (https://github.com/Gu-Lab-RBL-NCI/guide-passenger-FC), miRNA score calculation (https://github.com/Gu-Lab-RBL-NCI/miRNA-score-calculation) and cumulative distribution of miRNA target fold change (https://github.com/Gu-Lab-RBL-NCI/miRNA-target-FC).

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

## Acknowledgements

We thank R. Chari for designing and generating CRISPR reagents for the P19 HM cell line. This work was funded by the Intramural Research Program (ZIA-BC-011566) of the National Institutes of Health (NIH). The contributions of the NIH author were made as part of their official duties as NIH federal employees, are in compliance with agency policy requirements and are considered Works of the United States Government. However, the findings and conclusions presented in this paper are those of the authors and do not necessarily reflect the views of the NIH or the US Department of Health and Human Services.

## Author contributions

S.G., S.M. and A.Y. conceived and designed the study. S.M. carried out the experiments and data analyses with the help of C.S., A.Y., I.M., K.S., H.B. and W.G. S.M. and S.G. interpreted the data and wrote the paper.

## Competing interests

The authors declare no competing interests.

## Additional information

**Extended data** is available for this paper at https://doi.org/10.1038/s41594-025-01671-w.

**Correspondence and requests for materials** should be addressed to Shuo Gu.

**a**

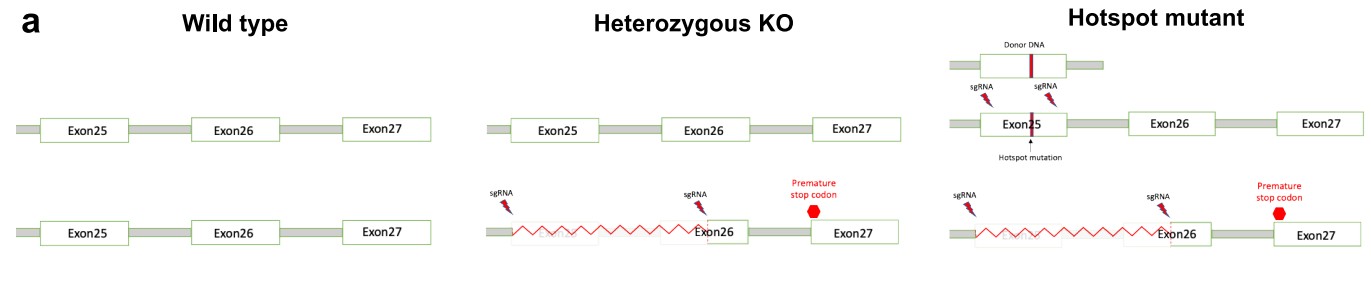

**b**

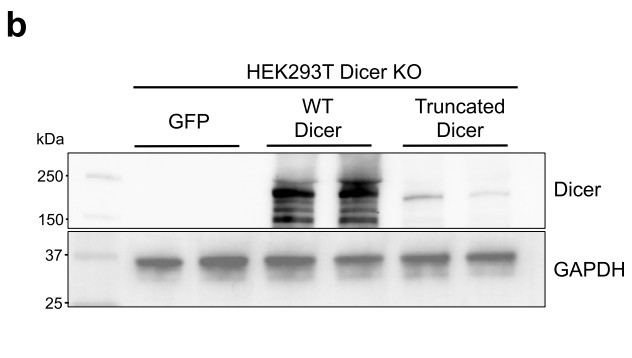

**c**

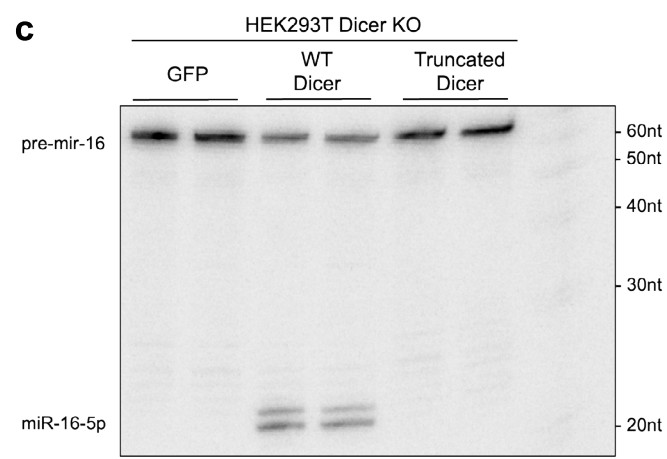

**d**

**e**

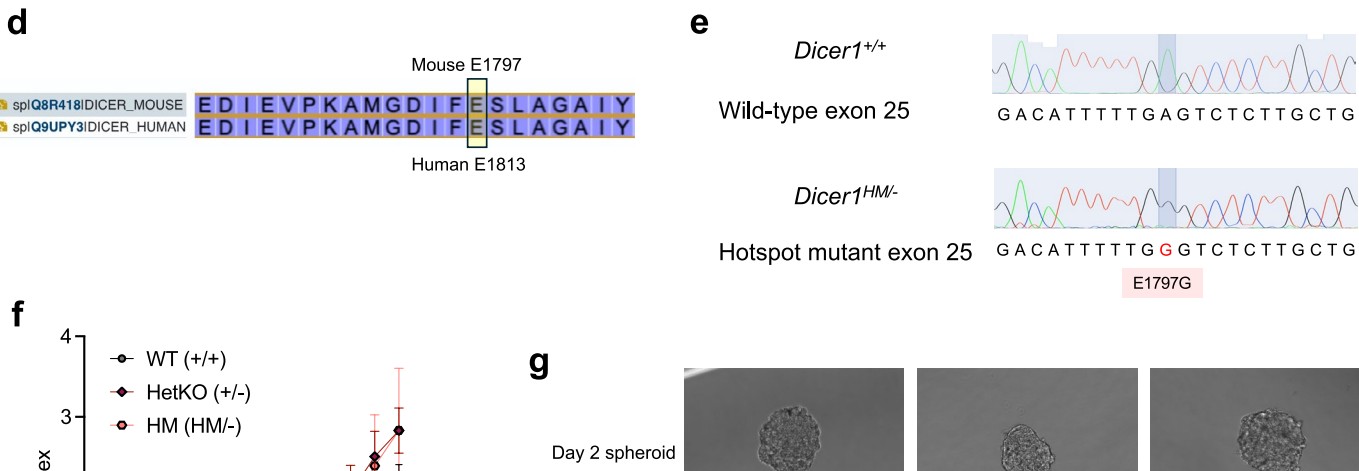

**f**

**g**

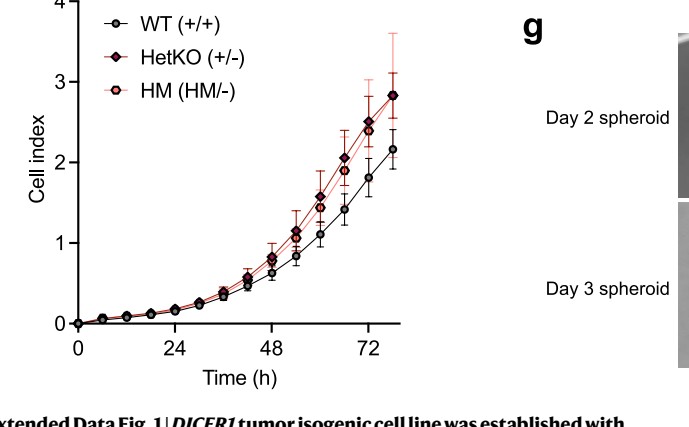

WT (+/+)          HetKO (+/-)          HM (HM/-)

**Extended Data Fig. 1 | *DICER1* tumor isogenic cell line was established with the P19 embryonic carcinoma cell line. a**) Illustration of sgRNAs and donor DNA used for CRISPR modification of P19 WT cells to generate HetKO and HM cells. **b**) Western blot for overexpression of GFP, WT Dicer, and truncated Dicer (mimicking the truncated allele of HetKO) in HEK293T Dicer KO cells. **c**) Northern blot for overexpression of GFP, WT Dicer, and truncated Dicer corresponding to the western blot in (b). **d**) Protein sequence alignment for human and mouse

Dicer for the E1813 hotspot mutation in *DICER1*-associated tumors using UniProt. **e**) Sanger sequencing chromatogram for the Dicer hotspot mutation in mouse exon 25 for HM compared to WT cells. **f**) Cell growth curve for WT, HetKO, and HM cells quantified as cell index by xCELLigence real-time cell analysis. **g**) Representative images of spheroids formed by WT, HetKO, and HM cells during growth in suspension using the hanging drop method.

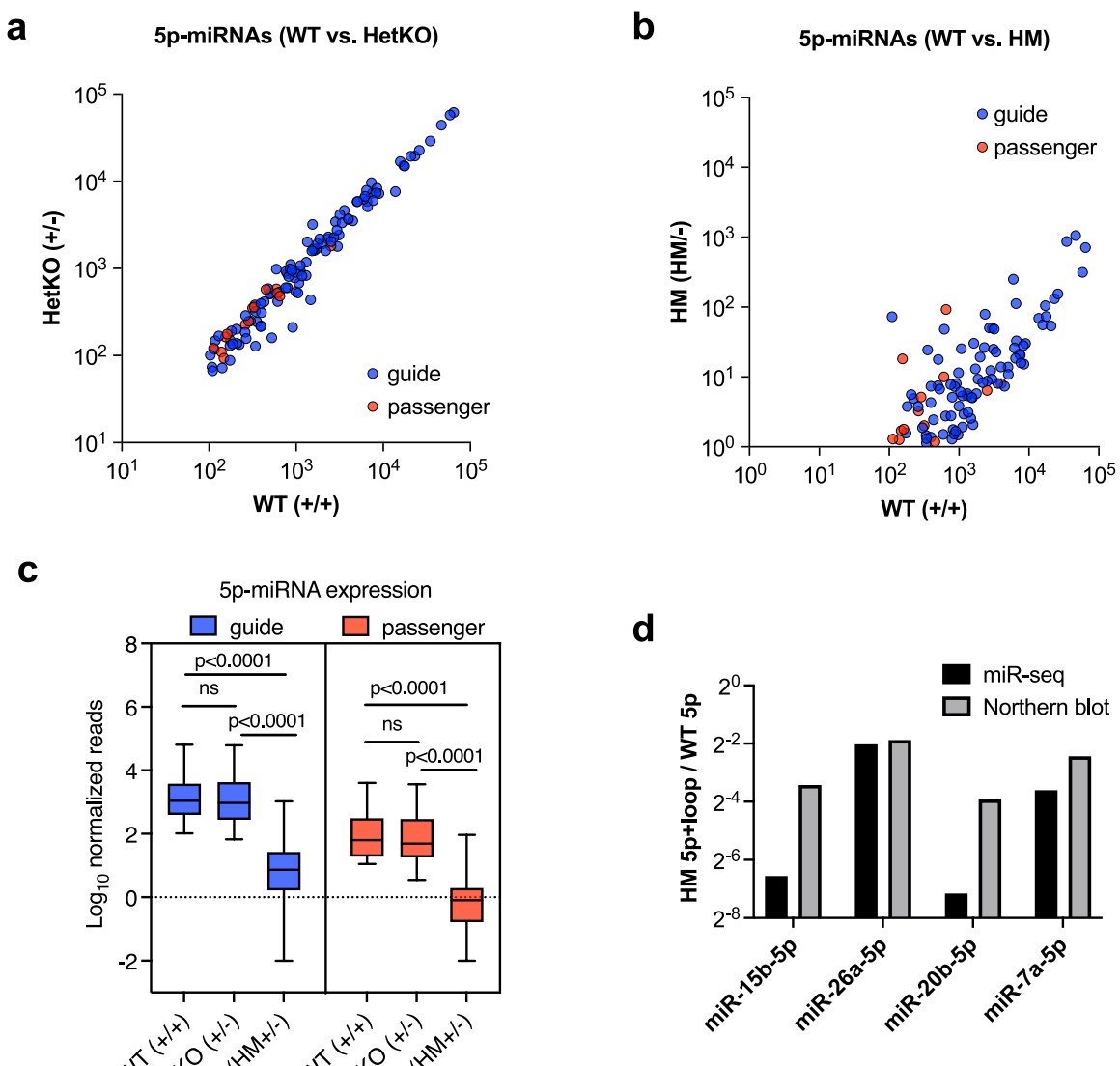

**Extended Data Fig. 2 | Expression of 5p-miRNAs is abolished in HM cells.**
**a)** Scatter plot of normalized reads of 5p-miRNA for WT vs. HetKO and **b)** WT vs. HM. Guide (blue) and passenger (red) miRNAs are classified based on relative expression to their complementary miRNA, where guide is the predominant strand of a mature miRNA duplex based on abundance ( > 70%) in the WT cells. **c)** Boxplot of miRNA expression shown as normalized reads for guide (n = 102)

and passenger (n = 60) 5p-miRNAs in each cell line. Average reads from n = 6 replicates per cell line were used. Boxplot shows the median (center line), 25th-75th percentile (box), and minimum to maximum values (whiskers). **d)** Barchart indicating the relative level of 5p-miRNA+loop in HM as detected in miR-seq and northern blot, normalized to 5p-miRNA in WT cells.

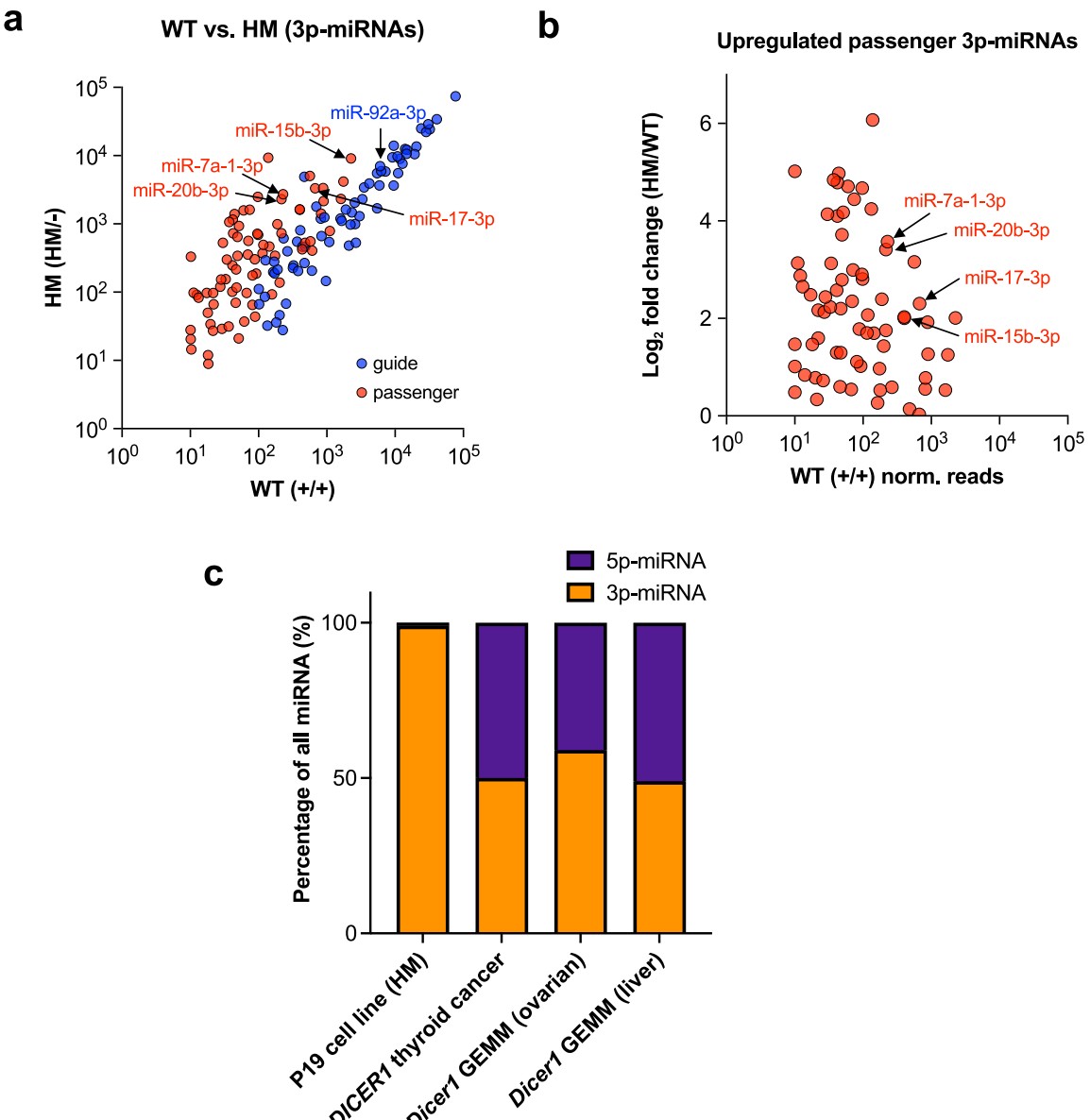

**Extended Data Fig. 3 | A subset of passenger 3p-miRNAs is upregulated in HM cells. a**) Scatter plot of normalized reads of guide (blue dots) and passenger (red dots) 3p-miRNAs for WT vs. HM. **b**) Scatter plot of normalized reads of upregulated passenger 3p-miRNA in WT vs. fold change in HM. **c**) Percentage of 5p- and 3p-miRNAs in HM cells, *DICER1* thyroid cancer, and *Dicer1* GEMM (ovarian and liver).

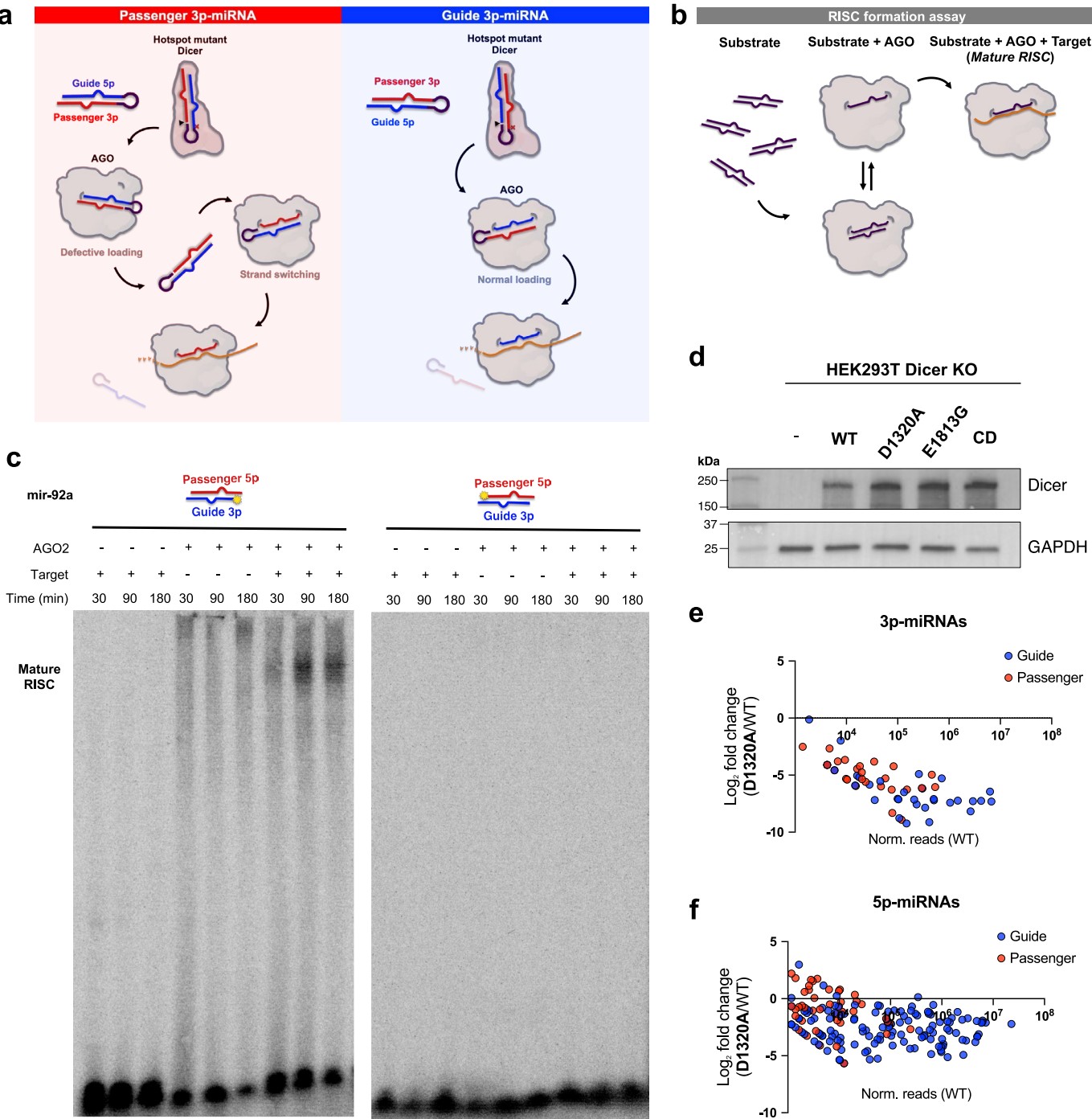

**Extended Data Fig. 4 | Dysregulation of AGO-loading results in passenger 3p-miRNA gain-of-function. a**) Illustration of differential AGO-loading for guide and passenger 3p-miRNAs. **b**) Illustration depicting mature RISC formation during RISC formation assay upon addition of AGO and target to the miRNA substrate. **c**) RISC formation assay for mir-92a in duplex and nicked precursor to visualize mature RISC formation. Star on duplex and nicked precursor illustration indicates the radiolabeled strand. Mature RISC formation is only observed when all components are present (substrate + AGO + target). **d**) Western blot validation for *DICER1* overexpression in HEK293T Dicer KO cells for WT, RNase IIIa mutant (D1320A), RNase IIIb mutant (E1813G), and catalytically dead (CD) Dicer. **e**) MA plot of differential expression of guide and passenger 3p-miRNAs and **f**) 5p-miRNAs in HEK293T Dicer KO cells rescued with RNase IIIa mutant (D1320A) vs. WT Dicer. Only miRNAs expressed in both WT and D1320A overexpression are shown.

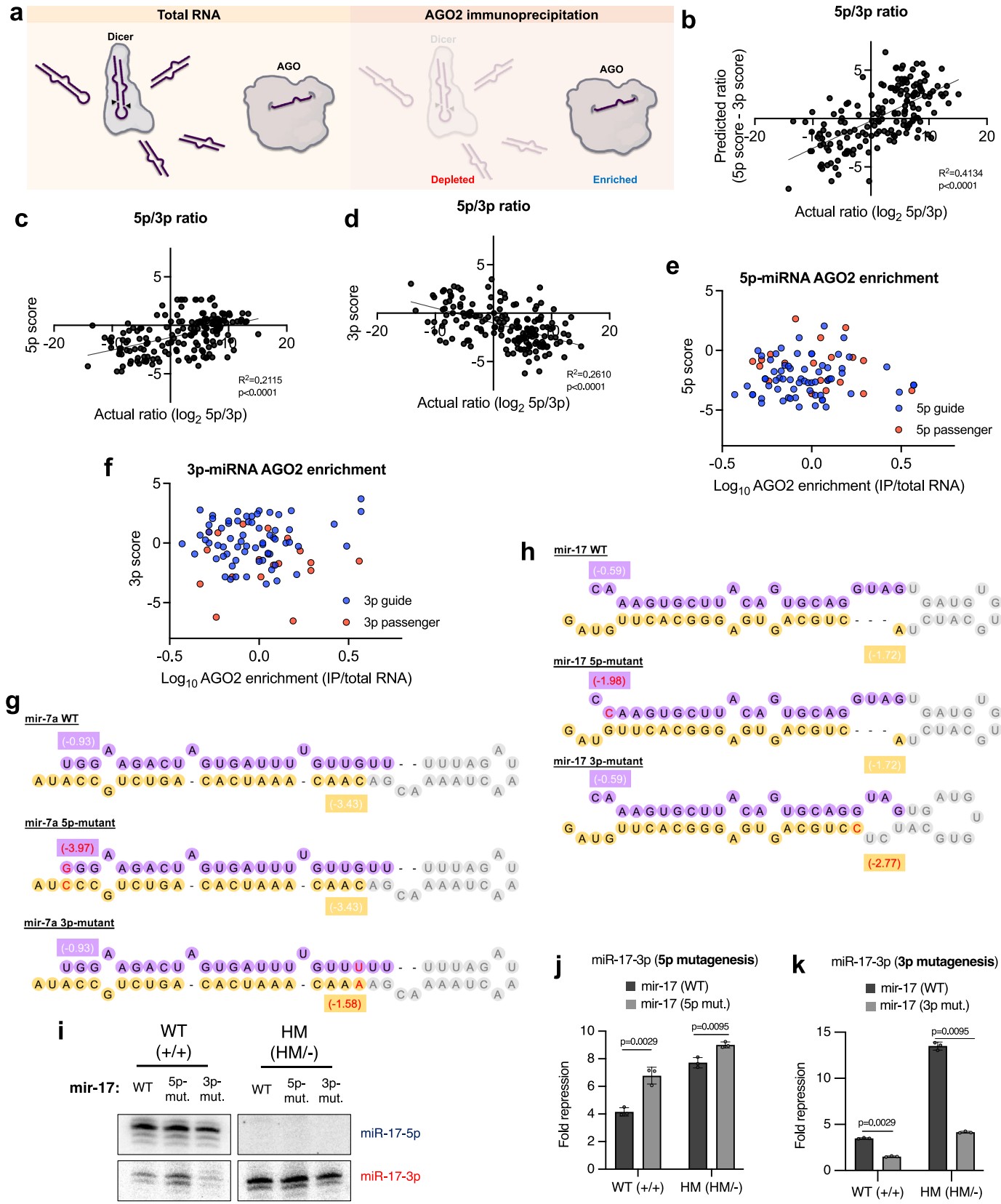

**Extended Data Fig. 5 | See next page for caption.**

**Extended Data Fig. 5 | The 5′ end features of upregulated passenger 3p-miRNAs determine their gain-of-function. a**) Illustration summarizing the enrichment of AGO-bound functional miRNA strand in AGO2-IP and depletion of other forms of miRNAs present in total RNA. **b**) Plot of 5p/3p ratio in P19 WT cells vs. predicted ratio, measured by the delta of 5p score and 3p score, or vs. individual scores (**c, d**). **e**) Plot of 5p-miRNAs and **f**) 3p-miRNA AGO2 enrichment (AGO2-IP/total RNA) vs. 5p- and 3p-score in WT cells, respectively. **g**) Illustration of mutagenesis design for mir-7a. Mutation for 5p-strand was done on the first nucleotide (U > G) to decrease the 5p-score and for 3p-strand (C > A) to increase the 3p-score. The scores before and after the mutation is shown in parenthesis highlighted in red. **h**) Illustration of mutagenesis design for mir-17. Mutation for 5p-strand was done on the second nucleotide (A > C) to decrease the 5p-score and for 3p-strand (A > C) to decrease the 3p-score. The scores before and after the mutation is shown in parenthesis highlighted in red. **i**) Northern blot for mir-17 mutagenesis. The 5p-strand was mutated to lower the 5p-score while 3p-strand was mutated to decrease the 3p-score. **j**) Evaluation of miR-17-3p function by reporter assay for 5p mutagenesis and (**k**) 3p mutagenesis corresponding to the northern blot in (**i**). Fold repression was calculated by normalizing to the seed mutant reporter for each miRNA. A two-sided Welch's t-test was used to calculate the p-values. Data shown as mean ± standard deviation.

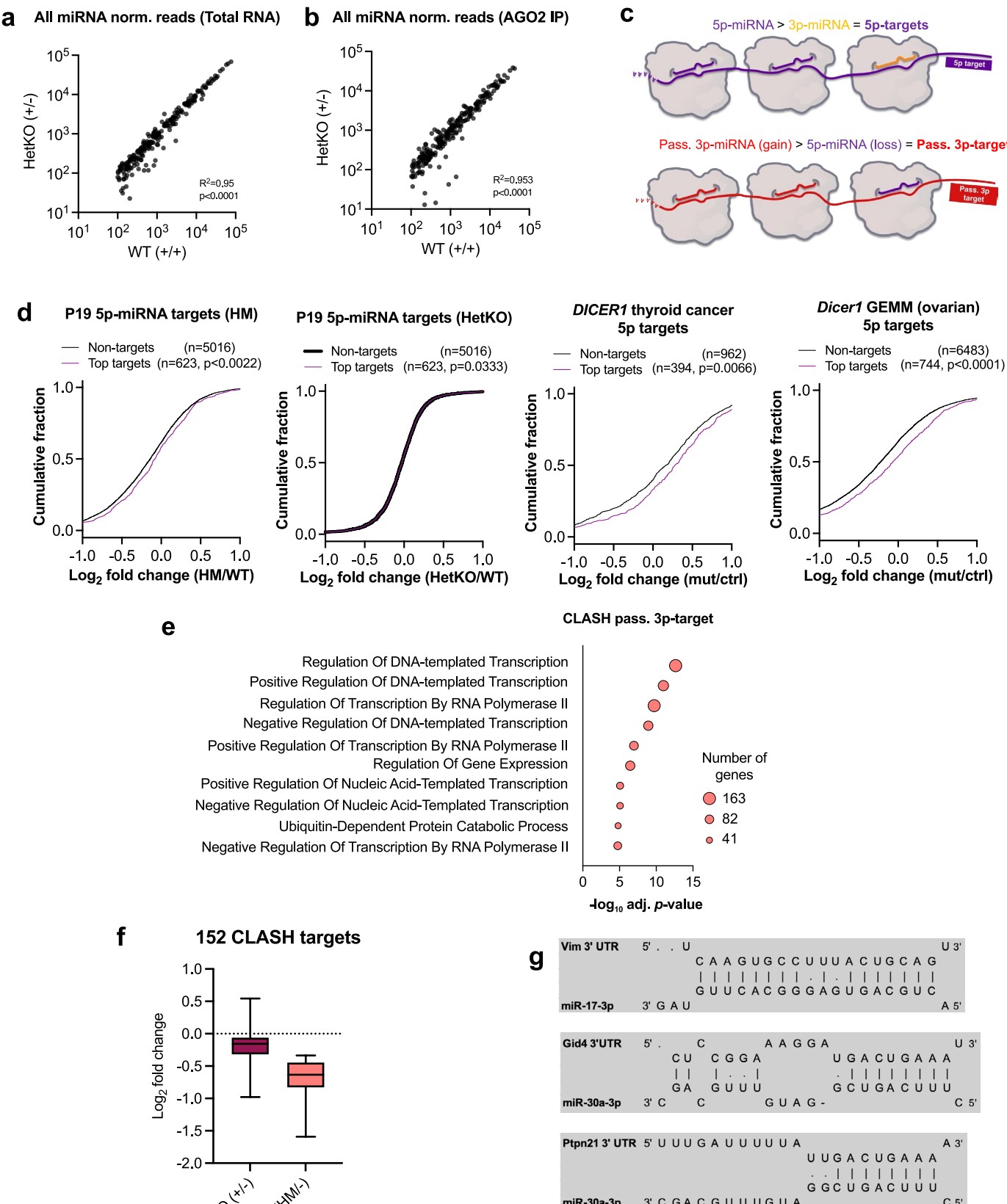

**Extended Data Fig. 6 | See next page for caption.**

**Extended Data Fig. 6 | Upregulated passenger 3p-miRNAs have functional impact on transcriptome. a**) Scatter plot of normalized reads of all miRNAs (n = 230) for WT vs. HetKO from total RNA miR-seq and **b**) AGO2-IP miR-seq. **c**) Illustration summarizing classification of 5p- and passenger 3p-targets for target prediction analysis. **d**) Cumulative distribution for fold change of top predicted 5p-targets in P19 cells (HM/WT and HetKO/WT), *Dicer1* ovarian GEMM (mut/ctrl), and *DICER1* thyroid cancer (mut/ctrl) datasets. Thicker line used for P19 HetKO to emphasize overlapping lines. A one-sided Kolmogorov-Smirnov test was used to calculate the p-values. **e**) Gene ontology analysis for all passenger 3p-targets from CLASH hybrids (n = 880). **f**) Boxplot of fold changes of downregulated CLASH passenger 3p-miRNA targets (n = 152) in HetKO and HM vs. WT. **g**) Visualization of complementarity between passenger 3p-miRNAs and their target sites for whole 3' UTR reporter assay.

# Reporting Summary

## Statistics

For all statistical analyses, confirm that the following items are present in the figure legend, table legend, main text, or Methods section.

| n/a | Confirmed | |
|---|---|---|
| ☐ | ☒ | The exact sample size (*n*) for each experimental group/condition, given as a discrete number and unit of measurement |
| ☐ | ☒ | A statement on whether measurements were taken from distinct samples or whether the same sample was measured repeatedly |
| ☐ | ☒ | The statistical test(s) used AND whether they are one- or two-sided<br>*Only common tests should be described solely by name; describe more complex techniques in the Methods section.* |
| ☒ | ☐ | A description of all covariates tested |
| ☐ | ☒ | A description of any assumptions or corrections, such as tests of normality and adjustment for multiple comparisons |
| ☐ | ☒ | A full description of the statistical parameters including central tendency (e.g. means) or other basic estimates (e.g. regression coefficient) AND variation (e.g. standard deviation) or associated estimates of uncertainty (e.g. confidence intervals) |
| ☐ | ☒ | For null hypothesis testing, the test statistic (e.g. *F*, *t*, *r*) with confidence intervals, effect sizes, degrees of freedom and *P* value noted<br>*Give P values as exact values whenever suitable.* |
| ☒ | ☐ | For Bayesian analysis, information on the choice of priors and Markov chain Monte Carlo settings |
| ☒ | ☐ | For hierarchical and complex designs, identification of the appropriate level for tests and full reporting of outcomes |
| ☐ | ☒ | Estimates of effect sizes (e.g. Cohen's *d*, Pearson's *r*), indicating how they were calculated |

*Our web collection on statistics for biologists contains articles on many of the points above.*

## Software and code

Policy information about availability of computer code

| Data collection | miRNA-seq and CLASH datasets were generated by Illumina NextSeq while  RNA-seq dataset was generated by Illumina NovaSeq. |
|---|---|
| Data analysis | Custom scripts are available on GitHub for cumulative distribution of guide/passenger 3p-miRNA fold change (https://github.com/Gu-Lab-RBL-NCI/guide-passenger-FC), miRNA score calculation (https://github.com/Gu-Lab-RBL-NCI/miRNA-score-calculation), and cumulative distribution of miRNA target fold change (https://github.com/Gu-Lab-RBL-NCI/miRNA-target-FC). |

For manuscripts utilizing custom algorithms or software that are central to the research but not yet described in published literature, software must be made available to editors and reviewers. We strongly encourage code deposition in a community repository (e.g. GitHub). See the Nature Portfolio guidelines for submitting code & software for further information.

## Data

Policy information about availability of data

All manuscripts must include a data availability statement. This statement should provide the following information, where applicable:
- Accession codes, unique identifiers, or web links for publicly available datasets
- A description of any restrictions on data availability
- For clinical datasets or third party data, please ensure that the statement adheres to our policy

The sequencing datasets generated in this study (miR-seq, RNA-seq, and CLASH) are deposited in the Gene Expression Omnibus (GEO) under accession codes

GSE279556, and GSE279645, and GSE279636. Source data can be accessed via Figshare at DOI: 10.6084/m9.figshare.29598575. All other data supporting the findings of this study are available from the corresponding author on reasonable request. Oligonucleotide sequences are provided as a supplementary data.

## Research involving human participants, their data, or biological material

Policy information about studies with human participants or human data. See also policy information about sex, gender (identity/presentation), and sexual orientation and race, ethnicity and racism.

| | |
|---|---|
| Reporting on sex and gender | *Use the terms sex (biological attribute) and gender (shaped by social and cultural circumstances) carefully in order to avoid confusing both terms. Indicate if findings apply to only one sex or gender; describe whether sex and gender were considered in study design; whether sex and/or gender was determined based on self-reporting or assigned and methods used.*<br>*Provide in the source data disaggregated sex and gender data, where this information has been collected, and if consent has been obtained for sharing of individual-level data; provide overall numbers in this Reporting Summary. Please state if this information has not been collected.*<br>*Report sex- and gender-based analyses where performed, justify reasons for lack of sex- and gender-based analysis.* |
| Reporting on race, ethnicity, or other socially relevant groupings | *Please specify the socially constructed or socially relevant categorization variable(s) used in your manuscript and explain why they were used. Please note that such variables should not be used as proxies for other socially constructed/relevant variables (for example, race or ethnicity should not be used as a proxy for socioeconomic status).*<br>*Provide clear definitions of the relevant terms used, how they were provided (by the participants/respondents, the researchers, or third parties), and the method(s) used to classify people into the different categories (e.g. self-report, census or administrative data, social media data, etc.)*<br>*Please provide details about how you controlled for confounding variables in your analyses.* |
| Population characteristics | *Describe the covariate-relevant population characteristics of the human research participants (e.g. age, genotypic information, past and current diagnosis and treatment categories). If you filled out the behavioural & social sciences study design questions and have nothing to add here, write "See above."* |
| Recruitment | *Describe how participants were recruited. Outline any potential self-selection bias or other biases that may be present and how these are likely to impact results.* |
| Ethics oversight | *Identify the organization(s) that approved the study protocol.* |

Note that full information on the approval of the study protocol must also be provided in the manuscript.

# Field-specific reporting

Please select the one below that is the best fit for your research. If you are not sure, read the appropriate sections before making your selection.

☒ Life sciences ☐ Behavioural & social sciences ☐ Ecological, evolutionary & environmental sciences

For a reference copy of the document with all sections, see nature.com/documents/nr-reporting-summary-flat.pdf

# Life sciences study design

All studies must disclose on these points even when the disclosure is negative.

| | |
|---|---|
| Sample size | No statistical method was used to predetermine sample size. Sample sizes were selected based on the magnitude of effects observed and previous literature. |
| Data exclusions | miRNAs expressed from a Chr12 cluster were excluded for Figures 1 - 5. The Chr12 miRNA cluster was not expressed in HM cell line due to unexpected silencing. Since the effect is transcriptional and a clonal effect, these miRNAs were excluded to avoid confounding the post-transcriptional effect on Dicer processing and AGO enrichment. |
| Replication | All attempts at replication were successful, and the number of replication is documented in the figure legends. |
| Randomization | Randomization is not relevant to this study. |
| Blinding | As data collection was instrument-based and quantitative, the risk of investigator bias was minimal. Each analysis was further supported by qualitative and orthogonal validation. |

# Reporting for specific materials, systems and methods

We require information from authors about some types of materials, experimental systems and methods used in many studies. Here, indicate whether each material, system or method listed is relevant to your study. If you are not sure if a list item applies to your research, read the appropriate section before selecting a response.

## Materials & experimental systems

| n/a | Involved in the study |
|---|---|
| ☐ | ☒ Antibodies |
| ☐ | ☒ Eukaryotic cell lines |
| ☒ | ☐ Palaeontology and archaeology |
| ☒ | ☐ Animals and other organisms |
| ☒ | ☐ Clinical data |
| ☒ | ☐ Dual use research of concern |
| ☒ | ☐ Plants |

## Methods

| n/a | Involved in the study |
|---|---|
| ☒ | ☐ ChIP-seq |
| ☒ | ☐ Flow cytometry |
| ☒ | ☐ MRI-based neuroimaging |

## Antibodies

| | |
|---|---|
| Antibodies used | Primary antibodies used in this study are rabbit anti-Dicer (1:1000, Novus Biological, NBP1-06520), mouse anti-α-tubulin (1:5000, Sigma, T9026), rabbit anti-GAPDH (1:1000, Cell Signaling Technology, 2118), and mouse anti-AGO2 (WAKO, 014-22023). |
| Validation | The rabbit anti-Dicer and mouse anti-AGO2 were validated for western blot and IP application according to Novus Biological and WAKO, respectively. Mouse anti-α-tubulin and rabbit anti-GAPDH were validated for western blot application according to Sigma and Cell Signaling Technology, respectively. All the antibodies used were validated for mouse specificity according to the respective manufacturers. |

## Eukaryotic cell lines

Policy information about cell lines and Sex and Gender in Research

| | |
|---|---|
| Cell line source(s) | P19 and HEK293T cell lines were obtained from ATCC. |
| Authentication | P19 cell line was not authenticated. HEK293T was authenticated by short-tandem repeat profiling. |
| Mycoplasma contamination | Cell lines were not tested for mycoplasma. |
| Commonly misidentified lines (See ICLAC register) | *Name any commonly misidentified cell lines used in the study and provide a rationale for their use.* |

## Plants

| | |
|---|---|
| Seed stocks | *Report on the source of all seed stocks or other plant material used. If applicable, state the seed stock centre and catalogue number. If plant specimens were collected from the field, describe the collection location, date and sampling procedures.* |
| Novel plant genotypes | *Describe the methods by which all novel plant genotypes were produced. This includes those generated by transgenic approaches, gene editing, chemical/radiation-based mutagenesis and hybridization. For transgenic lines, describe the transformation method, the number of independent lines analyzed and the generation upon which experiments were performed. For gene-edited lines, describe the editor used, the endogenous sequence targeted for editing, the targeting guide RNA sequence (if applicable) and how the editor was applied.* |
| Authentication | *Describe any authentication procedures for each seed stock used or novel genotype generated. Describe any experiments used to assess the effect of a mutation and, where applicable, how potential secondary effects (e.g. second site T-DNA insertions, mosiacism, off-target gene editing) were examined.* |

