## [Peer Review File · Nature Structural & Molecular Biology]

DICER1 Hotspot Mutation Induces 3p miRNA Gain of Function via Argonaute Strand Switch

Corresponding Author: Dr Shuo Gu

Version 0:

Decision Letter:

Nature Structural & Molecular Biology NSMB-PI49717

13th Sep 2024

Dear Dr. Gu,

Thank you for the presubmission inquiry regarding your manuscript "Pathogenic DICER1 RNase IIIb Hotspot Mutation Induces 3p miRNA Gain of Function via Argonaute Strand Switch". Based on the brief summary of your study, we are interested in considering your report for possible publication in Nature Structural & Molecular Biology. Please note that it is difficult to accurately assess a manuscript based on a brief description. Therefore, while we look forward to receiving your paper, we cannot guarantee that we would necessarily send the report out for review until we have read the entire manuscript.

In order to submit your complete manuscript, please use the link below:

Link Redacted

Sincerely,
Sara

Sara Osman, Ph.D.
Senior Editor
Nature Structural & Molecular Biology

Version 1:

Decision Letter:

20th Nov 2024

Dear Dr Gu,

Thank you again for submitting your manuscript "Pathogenic DICER1 RNase IIIb Hotspot Mutation Induces 3p miRNA Gain of Function via Argonaute Strand Switch". I am very sorry the process got so delayed and thank you again for your patience. We now have comments (below) from the 3 reviewers who evaluated your paper. In light of those reports, we remain interested in your study and would like to see your response to the comments of the referees in the form of a revised manuscript.

To guide the scope of the revisions, we list below a prioritized set of referee points that should be addressed in the revision, which we hope will be helpful to you. Please do not hesitate to contact me directly if you would like to discuss these issues further. Please be sure to address/respond to all concerns of the referees in full in a point-by-point response and highlight all changes in the revised manuscript text file.

We recommend dedicating efforts in revision to address the reviews as follows:

- Please address Rev#2's suggestions to strengthen the strand-switch conclusion (points #2-3; on a related note, please also address Rev#3's last point), address Rev#1's recommendation to further explore the role of the PAZ domain, provide additional analyses as requested by all revs and address technical points, requests for controls, discussion, and clarifications.

- We encourage you to consider Rev#3's comments about the functional studies and address them as possible.

Our revision period is 3 to 6 months. If you cannot send it within this time, please let us know. We will be happy to consider your revision as long as nothing similar has been accepted for publication at NSMB or published elsewhere. Should your manuscript be substantially delayed without notifying us in advance and your article is eventually published, the received date would be that of the revised, not the original, version.

Reporting Summary:

When submitting the revised version of your manuscript, please pay close attention to our [href="https://www.nature.com/nature-portfolio/editorial-policies/image-integrity">Digital Image Integrity Guidelines. and to the following points below:](https://www.nature.com/nature-portfolio/editorial-policies/image-integrity)

Please note that all key data shown in the main figures as cropped gels or blots should be presented in uncropped form, with molecular weight markers. These data can be aggregated into a single supplementary figure. While these data can be displayed in a relatively informal style, they must refer back to the relevant figures. These data should be submitted with the last revision, prior to acceptance, but you may want to start putting it together at this point.

We require deposition of coordinates (and, in the case of crystal structures, structure factors) into the Protein Data Bank with the designation of immediate release upon publication (HPUB). Electron microscopy-derived density maps and coordinate data must be deposited in EMDB and released upon publication. Deposition and immediate release of NMR chemical shift assignments are highly encouraged. Deposition of deep sequencing and microarray data is mandatory, and the datasets must be released prior to or upon publication. To avoid delays in publication, dataset accession numbers must be supplied with the final accepted manuscript and appropriate release dates must be indicated at the galley proof stage. Please find the complete NRG policies on data availability at <http://www.nature.com/authors/policies/availability.html>.

Nature Structural & Molecular Biology is committed to improving transparency in authorship. As part of our efforts in this direction, we are now requesting that all authors identified as 'corresponding author' on published papers create and link their Open Researcher and Contributor Identifier (ORCID) with their account on the Manuscript Tracking System (MTS), prior to acceptance. This applies to primary research papers only. ORCID helps the scientific community achieve unambiguous attribution of all scholarly contributions. You can create and link your ORCID from the home page of the MTS by clicking on 'Modify my Springer Nature account'. For more information please visit please visit

href="http://www.springernature.com/orcid">www.springernature.com/orcid.

Link Redacted

Sincerely,

Melina

Melina Casadio, PhD
Consulting Editor, Nature Structural & Molecular Biology
Senior Editor, Nature Cell Biology
ORCID ID: <https://orcid.org/0000-0003-2389-2243>

Referee expertise:

Referee #1: Argonaute, RISC, miRNAs

Referee #2: Argonaute, miRNAs, Dicer

Referee #3: Dicer/miRNAs in the context of cancer

Reviewers' Comments:

Reviewer #1 (Remarks to the Author):

Summary:

The Gu group developed an isogenic cell line from the P19 embryonic carcinoma cell line with a hotspot mutation in DICER1, termed the hotspot mutant (HM). They then conducted RNA-sequencing with spike-in normalization, along with northern blotting, to demonstrate that the RNase IIIb mutation in Dicer1 led to an increase in 3p-miRNAs while reducing 5p-miRNAs in HM cells. Biochemical assays further clarified the molecular mechanism, revealing that the 5p-loop was not selected as a guide by Ago2, which preferentially incorporated 3p-miRNAs into RISC complexes. Their data are highly robust, elegantly supporting their proposed model. This reviewer recommends publication in NSMB, provided the authors address the following requests.

Major Concerns:

The authors assessed endogenous and ectopic miR-17-3p for silencing Vimentin, one of the target genes. Since Fig. 6f identifies 152 promising targets, including additional targets regulated by 3p-miRNAs would further strengthen their conclusions.

The authors suggest that the PAZ domain may be involved in guide selection, proposing that the 5p-loop structure may hinder PAZ domain recognition. Do miRNAs require a specific length to be captured by the PAZ domain? Additionally, the Kay group previously reported that the PAZ domain is dispensable for AGO loading in slicing-competent RISCs. Could this finding conflict with the current study's conclusions?

Minor Concerns:

Figure 1d: Quantifying the bands would be helpful, as the authors indicate that the HM has more than half of WT expression.

Figure 1f: The labels for WT, HetKO, and HM are difficult to distinguish. Color coding would improve clarity.

Line 184: Including both Fig. 3b and Fig. 3d together in the citation would improve clarity.

Line 216: Data for HeKo cells seem to be missing from Fig. 4b.

Lines 277–278: It appears that Fig. 5c and Fig. 5d may be mislabeled or mixed up.

Line 333: A brief description of Vimentin's physiological role and its relevance to disease would enhance the discussion.

Reviewer #2 (Remarks to the Author):

Sharan Malagobadan et al. investigated how pathogenic hotspot mutations in DICER1 influence cancer progression by disrupting the miRNA biogenesis pathway. Using CRISPR, they introduced hotspot mutations into the RNase IIIb domain of the endogenous Dicer1 locus in a mouse embryonic carcinoma cell line. This allowed them to demonstrate that the hotspot mutations lead to a loss of 5p-miRNAs and a selective retention of certain 3p-miRNAs, resulting in a "strand-switch" phenomenon. Furthermore, they confirmed that this dysregulation is a direct consequence of the hotspot mutations, which may contribute to DICER1-associated tumorigenesis.

This study is conducted with precision and thoroughness. I believe it effectively elucidates how hotspot mutations in DICER1 impact miRNA expression and downstream genes, bringing us closer to understanding the mechanisms of oncogenesis driven by these mutations.

Major points:

- (1) The miRNA-seq analysis in this paper needs further detail. Even a single nucleotide shift at the miRNA's 5' end can alter the seed sequence, potentially affecting the target site significantly. However, the analysis lacks clarity, and it is uncertain how precisely the 5' end positions of miRNAs are conserved after Dicer cleavage. Additionally, miRNA activity may depend on whether wild-type Dicer and the hotspot-mutant Dicer maintain similar cleavage accuracy or if the mutant Dicer's cleavage site is misaligned. These aspects are not addressed in the paper.
- (2) The hotspot mutation in Dicer makes 3p-miRNA functional by preventing the 5p+loop sequence from effectively incorporating into the Ago protein. This effect is especially notable when the 5p-miRNA initially serves as the guide strand. The absence of 5p-miRNA uptake by Ago allows the 3p-miRNA, originally the passenger strand, to take over the guide function, significantly affecting 3p-miRNA. This phenomenon may be due to the loop portion of the 5p+loop inhibiting Ago loading. To clarify, the incorporation of 5p+loop versus 5p sequences into Ago should be compared.
- (3) The authors demonstrate that the enrichment of 3p-miRNAs into Ago correlates with 3p-scores rather than 5p-scores. Conversely, they should also examine if 5p-miRNA enrichment to Ago correlates with 5p-scores rather than 3p-scores. Additionally, exploring whether the same principle applies to 5p+loop sequences could clarify the reasons behind the loss of 5p+loops.

Minor points:

- (1) Figure 1f: The figure contains four asterisks, but the legend does not provide an explanation for them.
- (2) Figure 2a: Similarly, four asterisks are present in the figure without any corresponding explanation in the legend.
- (3) Figure 2b: The marks for "3p-miRNAs" and "Others" are difficult to distinguish, making it challenging to interpret the data accurately.

Reviewer #3 (Remarks to the Author):

Here Malagobadan et al. show that the loss of DICER1 RNase IIIb activity results in the downregulation of 5p miRNA processing and novel strand selection switching of passenger 3p miRNAs into guide 3p miRNAs. They generate DICER1 +/- ("HetKO") and DICER1E1797K/- ("HM") P19 embryonic carcinoma lines to model both heterozygous DICER1 loss and heterozygous DICER1 loss with second somatic hit in the RNase IIIb domain as frequently observed in DICER1 tumor predisposition patients. miRNA-seq and Northern blots revealed that in HM cells 5p miRNA processing was abolished, and that many 3p miRNAs typically designated as "passenger" miRNAs in DICER1WT cells were upregulated. Sequencing in human DICER1-associated thyroid tumors and two GEMMS of DICER1 tumor predisposition cancers (all harboring RNase IIIb domain mutations) demonstrated a similar miRNA phenotype in vivo. Dicer- and AGO-CLIP showed that several 3p upregulated passenger miRNAs were indeed loaded onto AGO and not sequestered by Dicer. Luciferase reporter assays confirmed that these "passenger-loaded" AGO species could target 3' UTR and repress target genes. Furthermore, the authors modeled the miRNA product of DICER1E1797K/- by supplying a 3p nicked pre-miRNA where the 5p strand is typically the guide and found that the radiolabeled 3p passenger strands were preferentially incorporated into the mature RISC when a 3p nicked pre-miRNA is supplied. Experiments modulating the 3p or 5p score via mutagenesis suggested that strand score still plays a key role in determining strand selection even in the setting of DICER1E1797K/-. RNA-seq and AGO-CLASH illustrated that 5p targets were de-repressed in HM cells and that upregulated 3p miRNAs were targeting the RISC to predicted 3' UTR targets, resulting in their repression.

This study generates an exciting model system that shows functional 3p passenger strand upregulation in HM cells which is supported by sequencing in vivo. While the miRNA strand selection phenotype appears robust, it remains unclear how this supports tumorigenesis or promotion. There is some evidence presented that differentiation is impaired in HM cells but there is no increase in doubling time.

Major Comments:

The authors mention a caveat that P19 cells may not serve as a cell of origin for DICER1 tumor predisposition tumors. Many studies have illustrated that DMSO treatment of P19 cells can cause myogenic differentiation of these cells (McBurney, M.W., et al, Nature 1982, 299:165). Therefore, the P19 cells are poised to differentiate into a potential cell of origin of rhabdomyosarcoma, a known DICER1-associated tumor. It is striking that the DMSO-induced differentiation did not induce mesoderm by the markers tested. Were any changes in growth/tumorigenesis observed in the HM cells after DMSO differentiation? Similarly, do HM cells develop tumors faster than DICER1WT P19 cells in vivo? More discussion about the

potential link between this mutation and the intersection with cell biology to result in transformation. These questions could be adequately addressed in the discussion and part of future studies.

It has been well illustrated that loss of one copy of Dicer1 functions as a haploinsufficient tumor suppressor. This has been shown in several genetically-engineered mouse models (Kumar et al, Genes and Dev, 2009, 23:2700; Lambertz, I., 2010, 17:633; Zindy, F., 2015, 10:e0129642). The statement on line 153-154 regarding losing a single Dicer allele not being pathogenic is not entirely accurate and should be revised. Would this not suggest that the Dicer1 +/- cells rather than Dicer1-WT should be used as controls to explore the differences with CLASH-seq and RNA-seq? Additionally, the paired RNA-seq and AGO-CLASH dataset provide an opportunity to rigorously evaluate the newly acquired "targets" of repression in HM cells. How do these targets compare to de-repression in Dicer1 +/- cells? These results will show what signaling is unique in HM cells compared to HetKO cells and may reveal why this second somatic hit is often selected for in tumors.

Finally, the data shown in Fig 4 suggests that 3' pre-nicking of the pre-miRNA is sufficient to change the "fated" passenger strand into the guide strand. Is 5' nicking of the pre-miRNA strand sufficient to mediate strand switching of pre-miRNA species? If not, this could potentially provide rational for mutations in the RNase IIIb domain over the RNase IIIa domain in cancer.

Version 2:

Decision Letter:

Our ref: NSMB-A49717B

27th Jun 2025

Dear Dr. Gu,

Thank you for submitting your revised manuscript "Pathogenic DICER1 RNase IIIb Hotspot Mutation Induces 3p miRNA Gain of Function via Argonaute Strand Switch" (NSMB-A49717B). It has now been seen by the original referees and their comments are below. The reviewers find that the paper has improved in revision, and therefore we'll be happy in principle to publish it in Nature Structural & Molecular Biology, pending minor revisions to comply with our editorial and formatting guidelines.

Thank you again for your interest in Nature Structural & Molecular Biology. Please do not hesitate to contact me if you have any questions.

Sincerely,

Melina Casadio, PhD
Locum Chief Editor, Nature Structural & Molecular Biology
ORCID ID: <https://orcid.org/0000-0003-2389-2243>

Reviewer #1 (Remarks to the Author):

The author adequately addressed the requests from me and other reviewers. This referee accepts the revised version

Reviewer #2 (Remarks to the Author):

A. Summary of key results

This manuscript examines how pathogenic hotspot mutations in the RNase IIIb domain of DICER1 affect miRNA biogenesis and function. Using CRISPR to introduce mutations into the endogenous Dicer1 locus in mouse cells, the authors show a selective loss of 5p-miRNAs and retention of 3p-miRNAs, causing a strand-switch phenomenon. They provide mechanistic insight that the mutation impairs AGO loading of the 5p strand, allowing the 3p strand to become functional.

B. Originality and significance

The study offers novel mechanistic understanding of how DICER1 mutations affect miRNA strand selection and function. The use of endogenous genome editing and thorough biochemical analyses adds originality and rigor.

C. Data & methodology

The experimental design is appropriate and well-executed. The CRISPR model and biochemical assays, including AGO loading and miRNA-seq, are suitable.

D. Statistical analyses and data presentation are clear. Minor figure issues have been corrected.

E. Conclusions

The conclusions are well supported and present a convincing model for how RNase IIIb mutations alter miRNA strand usage. The authors acknowledge limitations regarding miRNA 5' end precision and 5p+loop quantification as future work.

F. Suggested improvements

Further detailed analysis of Dicer cleavage fidelity would enhance the study, but the authors said that it is beyond the current scope and should be noted in the discussion. Quantitative analysis of 5p+loop AGO loading is desirable but limited by technical challenges. Figure legends and clarity improvements have been addressed.

G. References

Relevant prior work is cited appropriately, situating the study within the existing literature.

H. Clarity and context

The manuscript is clearly written with a lucid abstract, introduction, and discussion. The authors have constructively addressed reviewer comments and improved presentation.

Overall assessment

This manuscript makes a significant contribution to understanding the molecular impact of pathogenic DICER1 mutations. With revisions, it is suitable for publication in Nature Structural & Molecular Biology.

Reviewer #3 (Remarks to the Author):

The authors have very nicely addressed my prior comments with either the addition of new rigorous data or appropriate added clarifications or discussion in the text. This is a rigorous study and would be of interest to the non-coding RNA and cancer biology communities.

Version 3:

Decision Letter:

12th Aug 2025

Dear Dr. Gu,

We are now happy to accept your revised paper "DICER1 Hotspot Mutation Induces 3p miRNA Gain of Function via Argonaute Strand Switch" for publication as an Article in Nature Structural & Molecular Biology.

Your paper will be published online soon after we receive proof corrections and will appear in print in the next available issue. You can find out your date of online publication by contacting the production team shortly after sending your proof corrections.

Authors may need to take specific actions to achieve compliance with funder and institutional open access mandates. If your research is supported by a funder that requires immediate open access (e.g. according to [Plan S principles](https://www.springernature.com/gp/open-science/plan-s-compliance) or the [NIH public access policy](https://www.springernature.com/gp/open-science/us-federal-agency-compliance)) then you should select the gold OA route, and we will direct you to the compliant route where possible. Because authors warrant under our subscription licensing terms that they haven't committed to licensing any version of their article under a licence inconsistent with the terms of our agreement – including the applicable embargo period – publication under the subscription model isn't suitable for authors whose funders require no embargo.

Sincerely,

Melina Casadio, PhD
Locum Chief Editor, Nature Structural & Molecular Biology
ORCID ID: <https://orcid.org/0000-0003-2389-2243>

We would like to sincerely thank all reviewers for carefully reading our manuscript and providing positive feedback and constructive suggestions. Please find our point-by-point responses below. Reviewer comments are shown in *italic font*, and our responses are provided in **blue**. Corresponding changes to the manuscript have been made and are marked in **red**.

Reviewer 1

Summary:

The Gu group developed an isogenic cell line from the P19 embryonic carcinoma cell line with a hotspot mutation in DICER1, termed the hotspot mutant (HM). They then conducted RNA-sequencing with spike-in normalization, along with northern blotting, to demonstrate that the RNase IIIb mutation in Dicer1 led to an increase in 3p-miRNAs while reducing 5p-miRNAs in HM cells. Biochemical assays further clarified the molecular mechanism, revealing that the 5p-loop was not selected as a guide by Ago2, which preferentially incorporated 3p-miRNAs into RISC complexes. Their data are highly robust, elegantly supporting their proposed model. This reviewer recommends publication in NSMB, provided the authors address the following requests.

Major Concerns:

1) *The authors assessed endogenous and ectopic miR-17-3p for silencing Vimentin, one of the target genes. Since Fig. 6f identifies 152 promising targets, including additional targets regulated by 3p-miRNAs would further strengthen their conclusions.*

We appreciate this great suggestion. In response, we validated two additional predicted targets, Ptpn21 and Gid4, which are regulated by the upregulated miR-30a-3p. These targets were identified in our combined CLASH and RNA-seq analyses (Fig. 6f), and their gain of repression was confirmed using 3' UTR reporter assays in WT and HM cells. The results are presented in Figure 6g and Figure S6g.

Figure 6g: Reporter assay for passenger 3p-targets of miR-17-3p (Vim) and miR-30a-3p (Gid4 and Ptpn21) in WT and HM cells (n = 3 replicates per cell line).

Figure S6g: Visualization of complementarity between passenger 3p-miRNAs and their target sites for whole 3' UTR reporter assay.

2) *The authors suggest that the PAZ domain may be involved in guide selection, proposing that the 5p-loop structure may hinder PAZ domain recognition. Do miRNAs require a specific length to be captured by the PAZ domain? Additionally, the Kay group previously reported that the PAZ domain is dispensable for AGO loading in slicer-competent RISCs. Could this finding conflict with the current study's conclusions?*

We appreciate this insightful point. To our knowledge, no study has precisely defined the minimum length required for a miRNA to dock into the PAZ domain. However, based on structural insights, we speculate that miRNAs shorter than ~20 nucleotides are unlikely to be stably anchored within the PAZ pocket. As noted in the Kay group study, the PAZ domain is essential for slicer-independent unwinding but dispensable for slicer-dependent unwinding. Since most miRNA duplexes are not substrates for AGO2 slicing due to internal mismatches and bulges, slicer-independent unwinding is likely the predominant pathway and thus requires PAZ binding and/or N-domain wedging, consistent with our proposed model.

Nonetheless, this raises an intriguing question about miRNAs that can be loaded in a slicer-dependent manner. For example, miR-451 is known to be loaded through AGO2 slicing. Based on our model, we would expect miR-451, despite being a 5p-miRNA, to be efficiently loaded into AGO2 in HM cells. Our preliminary results support this prediction. We are also testing whether other precursors with near-perfect stem structures, such as many shRNAs that are cleavable by Dicer and potentially amenable to AGO2 slicing, are similarly loaded in HM cells. These studies may ultimately inform the design of shRNAs that are specifically processed in HM cells and could be leveraged for therapeutic applications.

While this line of investigation is beyond the scope of the current study, it raises important mechanistic questions that merit further exploration. Accordingly, we have acknowledged this point in the revised Discussion as follows:

" Notably, the unwinding step can be bypassed when the passenger strand is cleaved by AGO slicing^{8,45}, which also circumvents the requirement for PAZ domain docking

during guide strand loading⁴⁵. It would be intriguing to investigate how miRNAs with sliceable passenger strands are loaded into AGO in HM cells. "

Minor Concerns:

Figure 1d: Quantifying the bands would be helpful, as the authors indicate that the HM has more than half of WT expression.

Thank you for the suggestion. We have now quantified each band and included the normalized Dicer expression level in WT cells in the revised Figure 1d.

Figure 1d: Evaluation of Dicer protein expression by western blot, with each cell line run as duplicates across two independent experiments. Band intensity is denoted below each band (black font), while the normalized intensity for Dicer is denoted in parenthesis (red font).

Figure 1f: The labels for WT, HetKO, and HM are difficult to distinguish. Color coding would improve clarity.

Thank you for pointing this out. We have updated Figure 1f by adding color-coded labels for WT (gray), HetKO (maroon), and HM (coral) to improve clarity and readability. For consistency, we have applied the same color scheme across all figures where applicable and revised previous figures as needed.

Figure 1f: RT-qPCR of markers of the three germ layers (shown in parenthesis) after induction of differentiation using the hanging drop method. Fold change was calculated relative to the pre-differentiation level for each cell line (n = 6 replicates per cell line).

Line 184: Including both Fig. 3b and Fig. 3d together in the citation would improve clarity.

Thank you for the suggestion. We have updated the citation to include both Figure 3b and Figure 3d to improve clarity.

Line 216: Data for HeKo cells seem to be missing from Fig. 4b.

Thank you for catching this. “HetKO” was mistakenly included in this sentence. For the functional analyses (miRNA and UTR reporter assays, CLASH, etc.), we focused on comparisons between HM and WT cells. This has been corrected in the revised manuscript.

Lines 277–278: It appears that Fig. 5c and Fig. 5d may be mislabeled or mixed up.

We apologize for the confusion. Upon review, we found that the panels were correctly labeled but may not have been clearly presented. We have revised Figure 5c and Figure 5d to improve clarity and ensure the labeling is more immediately understandable.

Line 333: A brief description of Vimentin’s physiological role and its relevance to disease would enhance the discussion.

Thank you for the thoughtful suggestion. We have added the following text to the Discussion section to highlight the physiological role of Vimentin and its disease relevance:

“Notably, one validated target of these 3p-miRNAs is Vimentin, a key cytoskeletal component and a hallmark of mesenchymal cells⁴¹. HM cells failed to express specific mesodermal markers upon DMSO-induced differentiation, indicating a potential defect in mesenchymal lineage formation, a cell type thought to be the potential cell of origin for DICER1-associated tumors⁴². Together, these findings suggest that the gain-of-function in passenger 3p-miRNAs may play a critical role in tumorigenesis.”

Reviewer #2

Sharan Malagobadan et al. investigated how pathogenic hotspot mutations in DICER1 influence cancer progression by disrupting the miRNA biogenesis pathway. Using CRISPR, they introduced hotspot mutations into the RNase IIIb domain of the endogenous Dicer1 locus in a mouse embryonic carcinoma cell line. This allowed them to demonstrate that the hotspot mutations lead to a loss of 5p-miRNAs and a selective retention of certain 3p-miRNAs, resulting in a “strand-switch” phenomenon. Furthermore, they confirmed that this dysregulation is a direct consequence of the hotspot mutations, which may contribute to DICER1-associated tumorigenesis. This study is conducted with precision and thoroughness. I believe it effectively

elucidates how hotspot mutations in DICER1 impact miRNA expression and downstream genes, bringing us closer to understanding the mechanisms of oncogenesis driven by these mutations.

Major points:

(1) The miRNA-seq analysis in this paper needs further detail. Even a single nucleotide shift at the miRNA's 5' end can alter the seed sequence, potentially affecting the target site significantly. However, the analysis lacks clarity, and it is uncertain how precisely the 5' end positions of miRNAs are conserved after Dicer cleavage. Additionally, miRNA activity may depend on whether wild-type Dicer and the hotspot-mutant Dicer maintain similar cleavage accuracy or if the mutant Dicer's cleavage site is misaligned. These aspects are not addressed in the paper.

We thank the reviewer for this insightful comment and fully agree with the importance of analyzing Dicer cleavage fidelity and its impact on miRNA 5' end precision, which has been a long-standing interest of our group. Our preliminary data indicate that the Dicer RNase IIIb hotspot mutation can indeed generate isomiRs with altered 5' ends, potentially modifying seed sequences and contributing to the gain-of-function effects observed in HM cells.

However, the mechanisms by which this mutation selectively affects the cleavage fidelity of certain pre-miRNAs remain unclear and likely involve complex structural and sequence-specific factors. We believe that a comprehensive investigation of this phenomenon falls beyond the scope of the current study but represents an exciting direction for future work. We are actively pursuing this line of research and hope to report our findings in a subsequent publication. We sincerely thank the reviewer for raising this important point and respectfully hope they agree that this mechanistic question is best addressed in a dedicated follow-up study.

(2) The hotspot mutation in Dicer makes 3p-miRNA functional by preventing the 5p+loop sequence from effectively incorporating into the Ago protein. This effect is especially notable when the 5p-miRNA initially serves as the guide strand. The absence of 5p-miRNA uptake by Ago allows the 3p-miRNA, originally the passenger strand, to take over the guide function, significantly affecting 3p-miRNA. This phenomenon may be due to the loop portion of the 5p+loop inhibiting Ago loading. To clarify, the incorporation of 5p+loop versus 5p sequences into Ago should be compared.

We thank the reviewer for this excellent summary, which aligns well with the model we proposed, that the loop structure attached to the 3' end of the 5p-miRNA hinders its incorporation into AGO, thereby enabling the 3p strand to assume guide function. Strong experimental support for this model is provided in Figure 4d, where we examined AGO loading of miR-7a using either a duplex or a nicked precursor as substrate.

As the reviewer noted, when the duplex was used, AGO preferentially loaded the 5p strand (Fig. 4d, gel ii vs. gel i). However, with the nicked precursor, where the 5p strand

is still attached to the terminal loop, loading of the 5p+loop strand was completely abolished (Fig. 4d, gel iv vs. gel ii). This result supports the idea that the presence of the loop impedes AGO loading of the 5p strand.

We apologize for not stating this point more clearly in the original manuscript and have revised the text following the reviewer's suggestion:

“In contrast, results differed for the miR-7a-1 passenger 3p strand when comparing duplex and nicked precursor substrates. With the duplex, AGO predominantly loaded the 5p strand (Fig. 4d, comparing gel ii to gel i). However, when the nicked precursor was used, 5p+loop strand loading was completely abolished (Fig. 4d, comparing gel iv to gel ii), supporting our hypothesis that the loop structure at the 3' end of the 5p strand hinders its loading into AGO. Instead, the 3p strand was now preferentially selected as the guide (Fig. 4d, gel iii).”

Figure 4d: miR-7a-1 (passenger 3p-miRNA) in duplex and nicked precursor forms to visualize the mature RISC formation. The star on the duplex and nicked precursor illustration indicates the radiolabeled strand.

(3) *The authors demonstrate that the enrichment of 3p-miRNAs into Ago correlates with 3p-scores rather than 5p-scores. Conversely, they should also examine if 5p-miRNA enrichment to Ago correlates with 5p-scores rather than 3p-scores. Additionally, exploring whether the same principle applies to 5p+loop sequences could clarify the reasons behind the loss of 5p+loops.*

We would like to clarify that in HM cells, 5p strands become 5p+loop products due to the absence of RNase IIIb-mediated cleavage. These loop-attached strands are no longer in competition with the 3p strands, as the loop structure obstructs their incorporation into AGO. As a result, AGO loading in HM cells depends primarily on features of the 3p-miRNA, which explains why AGO enrichment correlates with 3p-scores rather than 5p-scores.

Conversely, in WT cells, both the 5p and 3p strands are generated and compete for AGO loading, making them suitable for assessing the influence of both strand scores. In this context, our data show that neither 5p-score nor 3p-score alone significantly correlates with AGO enrichment of their respective strands. Instead, the difference

between 5p- and 3p-scores better predicts strand selection. These results have been added in Figures S5e and S5f, and the corresponding text has been revised as follows:

“Guide strand selection typically involves competition between the two strands of a miRNA duplex, with the strand possessing the relatively higher score being favored. Indeed, the difference between 5p- and 3p-scores (Fig. S5b), rather than either score alone (Fig. S5c and S5d), correlates most strongly with the 5p/3p ratio in WT cells. Moreover, neither the 5p-score nor the 3p-score alone correlates with AGO enrichment of their respective strands (Fig. S5e and S5f). In contrast, our model predicts that in HM cells, AGO loading depends solely on the features of the 3p strand, as the loop attached to the 5p strand prevents its loading into AGO.”

Figure S5e: Plot of 5p-miRNAs and f) 3p-miRNA AGO2 enrichment (AGO2-IP/total RNA) vs. 5p- and 3p-score in WT cells, respectively.

Regarding the reviewer’s second point, it is indeed an interesting hypothesis that higher 5p-scores might predict tighter binding between 5p+loop products and AGO, potentially explaining their accumulation in HM cells. However, we are currently limited by technical challenges in accurately quantifying 5p+loop products. As noted in the manuscript, we observed discrepancies between northern blot and NGS measurements of 5p+loop levels (Figs. 2d and S2d), likely due to secondary structures interfering with library construction for some miRNAs. This limitation prevents reliable systematic analysis of 5p+loop abundance by NGS.

For the limited set of 5p+loop products measured by northern blot, we did not observe a strong correlation between their abundance and 5p-scores. Therefore, while the idea is compelling, we believe further analysis would require methodological improvements and is beyond the scope of the current study.

Minor points:

(1) *Figure 1f: The figure contains four asterisks, but the legend does not provide an explanation for them.*

Thank you for pointing this out, and we apologize for the omission. We have now added the explanation for the asterisks in the figure legend: “**** indicates $p < 0.0001$.”

(2) *Figure 2a: Similarly, four asterisks are present in the figure without any corresponding explanation in the legend.*

Thank you. We have made the same correction to the legend for Figure 2a.

(3) Figure 2b: The marks for "3p-miRNAs" and "Others" are difficult to distinguish, making it challenging to interpret the data accurately.

Thank you for pointing this out. We have revised Figure 2b to improve visual clarity by using a bright yellow color to indicate the "Others" category in the pie charts, making it easier to distinguish from "3p-miRNAs."

Figure 2b: Breakdown of specific miRNA population in each cell line.

Reviewer 3

Here Malagobadan et al. show that the loss of DICER1 RNase IIIb activity results in the downregulation of 5p miRNA processing and novel strand selection switching of passenger 3p miRNAs into guide 3p miRNAs. They generate DICER1^{+/-} ("HetKO") and DICER1^{E1797K/-} ("HM") P19 embryonic carcinoma lines to model both heterozygous DICER1 loss and heterozygous DICER1 loss with second somatic hit in the RNase IIIb domain as frequently observed in DICER1 tumor predisposition patients. miRNA-seq and Northern blots revealed that in HM cells 5p miRNA processing was abolished, and that many 3p miRNAs typically designated as "passenger" miRNAs in DICER1^{WT} cells were upregulated. Sequencing in human DICER1-associated thyroid tumors and two GEMMS of DICER1 tumor predisposition cancers (all harboring RNase IIIb domain mutations) demonstrated a similar miRNA phenotype in vivo. Dicer- and AGO-CLIP showed that several 3p upregulated passenger miRNAs were indeed loaded onto AGO and not sequestered by Dicer. Luciferase reporter assays confirmed that these "passenger-loaded" AGO species could target 3' UTR and repress target genes. Furthermore, the authors modeled the miRNA product of DICER1^{E1797K/-} by supplying a 3p nicked pre-miRNA where the 5p strand is typically the guide and found that the radiolabeled 3p passenger strands were preferentially incorporated into the mature RISC when a 3p nicked pre-miRNA is supplied. Experiments modulating the 3p or 5p score via mutagenesis suggested that strand score still plays a key role in determining strand selection even in the setting of DICER1^{E1797K/-}. RNA-seq and AGO-CLASH illustrated that 5p targets were de-repressed in HM cells and that upregulated 3p miRNAs were targeting the RISC to predicted 3' UTR targets, resulting in their repression.

This study generates an exciting model system that shows functional 3p passenger strand upregulation in HM cells which is supported by sequencing in vivo. While the miRNA strand selection phenotype appears robust, it remains unclear how this supports tumorigenesis or promotion. There is some evidence presented that differentiation is impaired in HM cells but there is no increase in doubling time.

Major Comments:

The authors mention a caveat that P19 cells may not serve as a cell of origin for DICER1 tumor predisposition tumors. Many studies have illustrated that DMSO treatment of P19 cells can cause myogenic differentiation of these cells (McBurney, M.W., et al, Nature 1982, 299:165). Therefore, the P19 cells are poised to differentiate into a potential cell of origin of rhabdomyosarcoma, a known DICER1-associated tumor. It is striking that the DMSO-induced differentiation did not induce mesoderm by the markers tested. Were any changes in growth/tumorigenesis observed in the HM cells after DMSO differentiation? Similarly, do HM cells develop tumors faster than DICER1WT P19 cells in vivo? More discussion about the potential link between this mutation and the intersection with cell biology to result in transformation. These questions could be adequately addressed in the discussion and part of future studies.

We thank the reviewer for this insightful comment and would first like to apologize for the error in the manuscript, where the hotspot mutation was incorrectly written as **E1797K**. The correct mutation introduced in HM cells is **E1797G**, and this has been corrected in the revised manuscript.

We appreciate the reviewer's thoughtful suggestion regarding the relevance of P19 cells and their capacity for myogenic differentiation upon DMSO treatment. This highlights an important opportunity to explore how the DICER1 hotspot mutation may intersect with developmental pathways to drive transformation. While our current study focuses on molecular mechanisms of miRNA dysregulation, we agree that examining growth and tumorigenic potential after DMSO-induced myogenic differentiation would be highly informative. We are actively considering this direction for future studies, including potential collaborative efforts for in vivo evaluation.

As suggested, we have expanded the Discussion section to reflect these points. Specifically, we now state:

“P19 cells are capable of differentiating into muscle lineage cells⁴³, whose precursor (primitive mesenchymal cells) is considered a potential cell of origin for rhabdomyosarcoma⁴⁴, a known DICER1-associated tumor¹⁸. It would be intriguing to evaluate the growth and tumorigenic potential of this panel of isogenic P19 cells following DMSO-induced myogenic differentiation, both in vitro and in vivo.”

Additionally, to acknowledge the observed differentiation defect in HM cells, we also added the following:

“HM cells failed to express specific mesodermal markers upon DMSO-induced differentiation, indicating a potential defect in mesenchymal lineage formation, a cell type thought to be the potential cell of origin for DICER1-associated tumors⁴².”

It has been well illustrated that loss of one copy of Dicer1 functions as a haploinsufficient tumor suppressor. This has been shown in several genetically-engineered mouse models (Kumar et al, Genes and Dev, 2009, 23:2700; Lambertz, I., 2010, 17:633; Zindy, F., 2015, 10:e0129642). The statement on line 153-154 regarding losing a single Dicer allele not being pathogenic is not entirely accurate and should be revised. Would this not suggest that the Dicer1^{+/-} cells rather than Dicer1-WT should be used as controls to explore the differences with CLASH-seq and RNA-seq? Additionally, the paired RNA-seq and AGO-CLASH dataset provide an opportunity to rigorously evaluate the newly acquired “targets” of repression in HM cells. How do these targets compare to de-repression in Dicer1^{+/-} cells? These results will show what signaling is unique in HM cells compared to HetKO cells and may reveal why this second somatic hit is often selected for in tumors.

We thank the reviewer for this thoughtful and important comment. We found that the miRNA expression profiles are nearly identical between WT and HetKO cells, suggesting that loss of a single DICER1 allele does not significantly disrupt miRNA biogenesis. Nonetheless, we fully agree that our original statement was not accurate. In the revised manuscript, we have acknowledged the well-established role of DICER1 as a haploinsufficient tumor suppressor and cited the relevant literature as following:

“Either way, this result suggests that Dicer’s established role as a haploinsufficient tumor suppressor^{18,34-36} is likely independent of its function in miRNA biogenesis.”

We also appreciate the reviewer’s suggestion that Dicer1^{+/-} (HetKO) cells could serve as a more appropriate control for dissecting the gain-of-function effects observed in HM cells. While we agree in principle, the near-complete concordance in miRNA expression and AGO2-IP profiles between WT and HetKO cells (Fig. S6a and S6b) led us to use WT as the primary control. This allowed us to simplify comparisons and maintain consistency across the study. We have clarified this rationale in the revised text:

“Since miRNA expression (Fig. S6a) and AGO2-IP data (Fig. S6b) showed near-perfect concordance between WT and HetKO, we focused our functional analyses on comparisons between HM and WT.”

Figure S6a and S6b: Scatter plot of normalized reads of all miRNAs for WT vs. HetKO from total RNA miR-seq and b) AGO2-IP miR-seq.

In addition, we examined the expression of target genes identified through our paired CLASH and RNA-seq datasets. These analyses revealed that de-repression of predicted 5p targets (Fig. S6d) and repression of 3p targets (Fig. S6f) are specific to HM cells and not observed in HetKO cells, further supporting the notion that these are gain-of-function effects unique to the hotspot mutation. These results have been included in Figure S6d and a newly added panel, Figure S6f. The text has been revised accordingly as follows:

“Consistent with the observation that passenger 3p-miRNA upregulation occurs in HM but not HetKO cells, downregulation of their target genes was specific to HM cells (Fig. 6f and S6f).”

Figure S6d: Cumulative distribution for fold change of top predicted 5p-targets in P19 cells (HM/WT and HetKO/WT)

Figure S6f: Boxplot of fold changes of downregulated CLASH passenger 3p-miRNA targets in HetKO and HM vs. WT.

While the current study focuses on the mechanistic basis underlying the gain-of-function of 3p-miRNAs, we agree that a systematic comparison between HetKO and HM cells would be valuable for identifying specific signaling pathways responsible for tumorigenesis. We have acknowledged this important direction in the Discussion section:

“By comparing HetKO and HM cells, it would be possible to rigorously investigate the newly acquired targets of gain-of-function 3p-miRNAs and identify the specific pathways affected. Such studies are essential to fully elucidate the oncogenic consequences of RNase IIIb hotspot mutations and to understand why these mutations are often selected as the second somatic hit in tumors.”

Finally, the data shown in Fig 4 suggests that 3' pre-nicking of the pre-miRNA is sufficient to change the “fated” passenger strand into the guide strand. Is 5' nicking of the pre-miRNA strand sufficient to mediate strand switching of pre-miRNA species? If not, this could potentially provide rational for mutations in the RNase IIIb domain over the RNase IIIa domain in cancer.

We thank the reviewer for this excellent question, which inspired a significant set of new experiments. As suggested, we tested whether nicking the 5' arm of the pre-miRNA is sufficient to induce strand switching. To this end, we generated a Dicer RNase IIIa catalytic mutant (**D1320A**), which produces precursors with a nick on the 5' arm (Fig. 4e) and expressed it in a Dicer-null cell line. WT Dicer, the RNase IIIb mutant, and a catalytically inactive Dicer served as controls.

We evaluated the impact of the RNase IIIa mutant on miRNA biogenesis by Northern blot and small RNA sequencing. Consistent with previous reports, 3p strand production was abolished (Fig. 4f, S4e), likely because the loop attached to the 5' end of the 3p strand interferes with recognition by the AGO MID domain. Notably, we did **not** observe consistent upregulation of 5p passenger strands. Instead, most 5p strands, regardless of their typical role as guide or passenger, were downregulated (Fig. 4f, S4f).

These findings suggest that even when the loop is not covalently linked to the 5p strand, it still obstructs access to the 3' end, preventing efficient AGO loading. Thus, free access to both ends of the guide strand appears to be essential for AGO incorporation. In addition to reinforcing our mechanistic model, these results may also provide insight into why cancer-associated DICER1 mutations predominantly affect RNase IIIb rather than RNase IIIa, exactly as the reviewer hypothesized.

We have incorporated these new findings into the revised manuscript as follows:

“To further test our model, we sought to determine whether precursors with a nick on the opposite strand would trigger a similar strand-switching phenomenon and lead to upregulation of the opposite strand. To this end, we generated a Dicer RNase IIIa mutant (D1320A) that produces precursors with a nick on the 5' arm (Fig. 4e) and expressed it in a Dicer-null cell line. WT Dicer, the RNase IIIb mutant, and a catalytically inactive Dicer served as controls (Fig. S4d). The impact of this mutant on miRNA biogenesis was examined using Northern blotting and small RNA sequencing. Consistent with previous findings²³, production of 3p strands was completely abolished (Fig. 4f and S4e), likely because the loop structure attached to the 5' end of the 3p strand interferes with recognition by the AGO MID domain. Interestingly, we did not observe consistent upregulation of the 5p passenger strands (Fig. 4f and S4f). Instead, most 5p strands, regardless of whether they typically function as guide or passenger, were downregulated (Fig. 4f and S4f). This suggests that although the loop portion is not covalently attached to the 5p strand, it nonetheless obstructs accessibility to its 3' end. These results further support our hypothesis that AGO loading requires free access to both the 5' and 3' ends of the guide strand. Moreover, they may help explain why cancer-associated DICER1 mutations predominantly affect RNase IIIb activity rather than RNase IIIa.”

Figure 4e: Illustration of the different nicked precursors produced by RNase IIIa and IIIb.

Figure 4f: Northern blot of Dicer overexpression (WT, RNase IIIa mutant, RNase IIIb mutant, and catalytically dead) in HEK293T Dicer KO cells, probed for guide and passenger 5p- and 3p-miRNAs (non-specific bands denoted with *)

Fig. S4e MA plot of differential expression of 3p-miRNAs in HEK293T Dicer KO cells rescued with RNase IIIa mutant (D1320A) vs WT-Dicer. (Note, many 3p-miRNAs which were not detected in D1320A are not shown in the plot)

Fig. S4f MA plot of differential expression of 5p-miRNAs in HEK293T Dicer KO cells rescued with RNase IIIa mutant (D1320A) vs WT-Dicer.